# Chd4 remodels chromatin to control retinal cell type specification and lineage termination

Sujay Shah[1,2], Suma Medisetti[1,2], José Alex Lourenço Fernandes[1,2] and Pierre Mattar[1,2,*]

## ABSTRACT

During development, neural progenitor cells modify their output over time to produce different types of neurons and glia in chronological sequences. Epigenetic processes have been shown to regulate neural progenitor potential, but the underlying mechanisms are not well understood. Here, we generated retina-specific conditional mouse knockouts (cKOs) in the key nucleosome remodeller *Chd4*. *Chd4* cKOs overproduced early-born retinal ganglion and amacrine cells. Postnatally, later-born rod photoreceptors were drastically underproduced. Progenitors failed to differentiate into Müller glia on schedule and continued to proliferate beyond their normal developmental window. Next, to determine how Chd4 regulates the genome, we performed CUT&RUN-seq and ATAC-seq, revealing that genome accessibility was significantly increased at ~10,000 regulatory elements. Accordingly, multiplexed single-cell transcriptomics demonstrated that deletion of *Chd4* led to corresponding increases in transcription. These results suggest that Chd4 restricts the genome to repress progenitor identity and promote differentiation. Taken together, our data suggest that Chd4-dependent nucleosome remodelling plays a crucial role in the temporal transition that governs lineage termination, but does not regulate earlier temporal transitions.

KEY WORDS: Nucleosome remodelling, Neural progenitor, Retinal Development, Neurogenesis, Competence, NuRD complex, Gliogenesis, Photoreceptors

## INTRODUCTION

The assembly of neural circuits depends on the choreographed production of a constellation of different types of neurons and glia. Neural progenitors generate this cell type diversity in precise numbers and proportions on a tightly regulated developmental schedule. In virtually every lineage, progenitors progressively alter their output over developmental time. For example, most CNS progenitors initially make neurons, but then irreversibly switch to producing glia (Miller and Gauthier, 2007). Moreover, highly complex sequences of neurons are produced in many regions, including the developing retina and mammalian neocortex. Landmark studies in vertebrate and invertebrate systems have demonstrated that the ordered production of different neurons and glia depends in part upon temporal transitions in progenitor competence states (Holguera and Desplan, 2018).

The vertebrate retina is a classic experimental model with which to understand the cell intrinsic mechanisms that modify neurogenesis over time. The mature retina is composed of six different types of neurons and one glial cell, which are generated by multipotent retinal progenitor cells (RPCs) (Holt et al., 1988; Turner and Cepko, 1987; Wetts and Fraser, 1988) that exhibit distinct phases of competence. During the early phase, RPCs initially produce retinal ganglion cells (RGCs), horizontals, cones and amacrine cells. As the generation of these cell types peaks, rod photoreceptors begin to be produced. In rodents, RPCs undergo a competence transition perinatally. Rods are generated in peak numbers during early postnatal stages, and RPCs also continue to generate amacrines but completely lose their capacity to produce the other early-born cell types. At the end of development, RPCs instead produce rods, bipolar cells and, finally, Müller glia (Alexiades and Cepko, 1997; Carter-Dawson and LaVail, 1979; Clark et al., 2019; Rapaport et al., 2004; Young, 1985). RPCs also give rise to 'neurogenic' precursors that have more restricted proliferative and developmental potential. Retinal neurons are often thought to arise from specific neurogenic precursors, whereas Müller glia are thought to differentiate directly from RPCs without passing through a neurogenic intermediate (Lyu et al., 2021; Shiau et al., 2021). This ordered production of retinal cell types is conserved across all vertebrates.

While we know a great deal about the developmental mechanisms that program retinal cell type identities, we know much less about how RPC competence is controlled. Cell-extrinsic signals have been shown to regulate progenitor output, although these factors are mainly thought to act as post-hoc feedback systems that refine cell type production (Bassett and Wallace, 2012; Cayouette et al., 2003). Previous work has also identified candidate cell-intrinsic competence determinants, which include transcription factors and microRNAs (Clark et al., 2019; Elliott et al., 2008; Georgi and Reh, 2010; Gordon et al., 2013; La Torre et al., 2013; Liu et al., 2013, 2020; Mattar et al., 2015; Poche et al., 2008). In both the retina and other CNS regions, competence transitions have also been shown to depend on heterochromatic determinants, including DNA methylation and the polycomb repressor complex (Hirabayashi et al., 2009; Takizawa et al., 2001; Zhang et al., 2023). These observations suggest a model where heterochromatic processes might act downstream of transcription factors and microRNAs. The 'decommissioning' of genes associated with early competence could explain the progressive restriction of developmental potential observed during retinal development.

The nucleosome remodelling and deacetylase (NuRD) complex is a strong candidate for integrating dynamically expressed temporal

[1]Regenerative Medicine Program, Ottawa Hospital Research Institute (OHRI), Ottawa, ON K1H 8L6, Canada. [2]Department of Cellular & Molecular Medicine and Department of Ophthalmology, University of Ottawa, Ottawa, ON K1H 8M5, Canada.

*Author for correspondence (pmattar@ohri.ca)

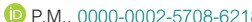 P.M., 0000-0002-5708-6218

transcription factors with heterochromatic effectors. NuRD has both histone deacetylation and nucleosome remodelling activities. In neural progenitors, the nucleosome remodelling activity is mainly provided by Chd4, a chromodomain helicase protein (Nitarska et al., 2016). Previous work in the cerebellum has shown that Chd4 is required to decommission genes and regulate higher-order genome looping (Goodman et al., 2020; Yamada et al., 2014; Yang et al., 2016). In the neocortex, precocious gliogenesis was also observed when the NuRD subunit *Mbd3* was mutated (Tsuboi et al., 2018), suggesting that NuRD might regulate the transition from neurogenic to gliogenic competence. Moreover, *Chd4* cKOs exhibited a loss in late-born upper-layer cortical neurons, suggesting that Chd4 might additionally regulate earlier competence transitions (Larrigan et al., 2023; Nitarska et al., 2016). Interestingly, the NuRD complex has been shown to interact with several temporal transcription factors linked to progenitor competence, including Ikzf1, Casz1 and Foxp1 (Chokas et al., 2010; Kehle et al., 1998; Kim et al., 1999; Liu et al., 2015; Mattar et al., 2021). However, the genetic requirement for Chd4 has not previously been examined in the context of retinal development.

Here, we used conditional genetics to study how Chd4 regulates retinal development. *Chd4* conditional knockout (cKO) mice markedly overproduced early-born RGCs, whereas late-born rod photoreceptors were decreased. RPCs failed to differentiate into Müller glia on schedule and dividing progenitors accumulated in the cKO. Surprisingly, despite the fact that neurogenesis was skewed towards early-born fates and away from later-born rods, we found that early RPC competence is not prolonged in the *Chd4* cKO. Thus, while RPCs depend upon Chd4 and nucleosome remodelling for the temporal transition that terminates the RPC

lineage at the end of retinal development, they do not require Chd4 for earlier temporal transitions. Taken together, these data hint that sequential competence transitions are regulated by different epigenetic mechanisms.

## RESULTS

### Chd4 expression during retinal development

We first utilized immunohistochemistry to examine the spatiotemporal expression profile of Chd4 during mouse retinal development. We found that Chd4 was ubiquitously expressed from embryonic day (E) 11.5 through to adult stages (Fig. 1). Expression levels were relatively constant within the nuclei of Ki67[+] RPCs from E13.5 through to postnatal day (P) 2 (Fig. 1D-K). However, Chd4 levels became somewhat elevated in postmitotic neurons within the ganglion cell layer (GCL). At P0 and P2, elevated expression was also apparent within postmitotic Ki67-negative cells within the outer neuroblastic layer (ONBL; Fig. 1G-K). In the adult retina, Chd4 expression levels remained particularly elevated within inner nuclear layer (INL) and GCL neurons. Weak expression was also apparent in rod photoreceptors (Fig. 1L-O). To address other (group II subfamily) Chd4 paralogs, we examined previously published retinal scRNA-seq data (Clark et al., 2019). While *Chd4* was expressed at high levels in RPCs, both *Chd3* and *Chd5* were mainly expressed in postmitotic neurons (Fig. S1). These results indicate that Chd4 is the main group II paralog in RPCs, but that its expression is not temporally dynamic.

### Chd4 cKO affects postnatal retinal histogenesis

Since *Chd4* is an essential gene (O'Shaughnessy-Kirwan et al., 2015), we next generated *Chd4* conditional knockouts (cKOs) using

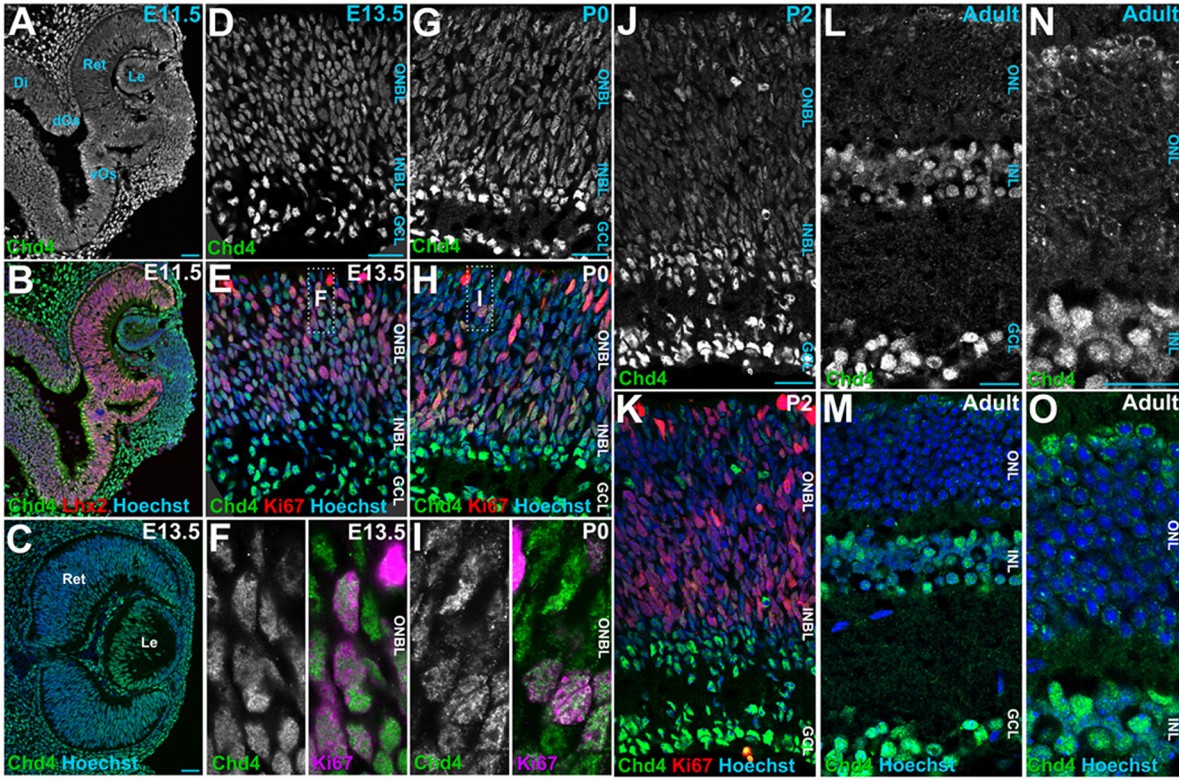

**Fig. 1. Chd4 expression dynamics during retinal development.** (A,B) E11.5 retinas stained for Chd4 and Lhx2. (C-F) Co-staining of Chd4 and Ki67 at E13.5 (D-F), P0 (G-I) and P2 (J,K). (L-O) Adult retinas stained for Chd4 and with Hoechst. Ret, retina; Le, lens; Di, diencephalon; dOs, dorsal optic stalk; vOs, ventral optic stalk; ONBL, outer neuroblastic layer; INBL, inner neuroblastic layer; ONL, outer nuclear layer; INL, inner nuclear layer; GCL, ganglion cell layer. Scale bars: 20 μm.

an allele with loxP sites flanking exons 12-21, which together encode the ATPase/helicase domain (Williams et al., 2004). This cassette was deleted using the *Chx10-Cre-GFP* driver (Rowan and Cepko, 2004), which expresses a Cre-GFP fusion protein in RPCs, beginning at ~E10.5. Towards the end of development, Cre-GFP expression is maintained in bipolars and weakly in some Müller glia. Full-length Chd4 protein (expected size 219 kDa) was efficiently abrogated in cKO retinas (Fig. 2A-C). Although the *Chx10-Cre-GFP* driver is prone to mosaicism, in our study we observed that an average of ~70% of cells within the ONBL expressed GFP in perinatal Cre+ animals (Fig. S2), which is similar to the overall proportions of RPCs within the layer (e.g. see Fig. 4).

Next, the effect of *Chd4* ablation was assessed at various stages between E16.5 and P15 (Fig. 2D-F, Fig. S3). *Chd4* cKOs exhibited a markedly expanded GCL along with a poorly formed inner plexiform layer. The distinct neuropil dividing the ONBL and GCL in the wild type and conditional heterozygote (chet) was missing in

the cKOs (Fig. 2D-F, Fig. S3). However, when the total number of cells was quantified between the three genotypes, no significant changes were observed at P0 or P2 (Fig. 2G,H; Fig. S3). This indicates that the expansion of the GCL does not arise as a consequence of significant retinal overgrowth, nor from premature differentiation of the progenitor pool.

At P8, *Chd4* cKOs exhibited disorganized retinal lamination. Along with the expanded GCL, mutant retinas exhibited a thinned ONL as compared to wild type or chets, and additionally exhibited ectopic GFP+ nuclei within the ONL (Fig. 2F). Again, when cells were quantified, no significant difference was observed between wild-type, chet, and cKO retinas (Fig. 2I). At P15 the loss of *Chd4* resulted in a hyperplastic ONL containing ectopic GFP+ cells (Fig. 2F, Fig. S4). cKO retinas exhibited an approximately 1.5-fold decrease in cell numbers when compared to wild type or chet (Fig. 2J). These data suggest that, in the cKO, cell death significantly reduces cell numbers between P8 and P15 (Fig. 2K), likely

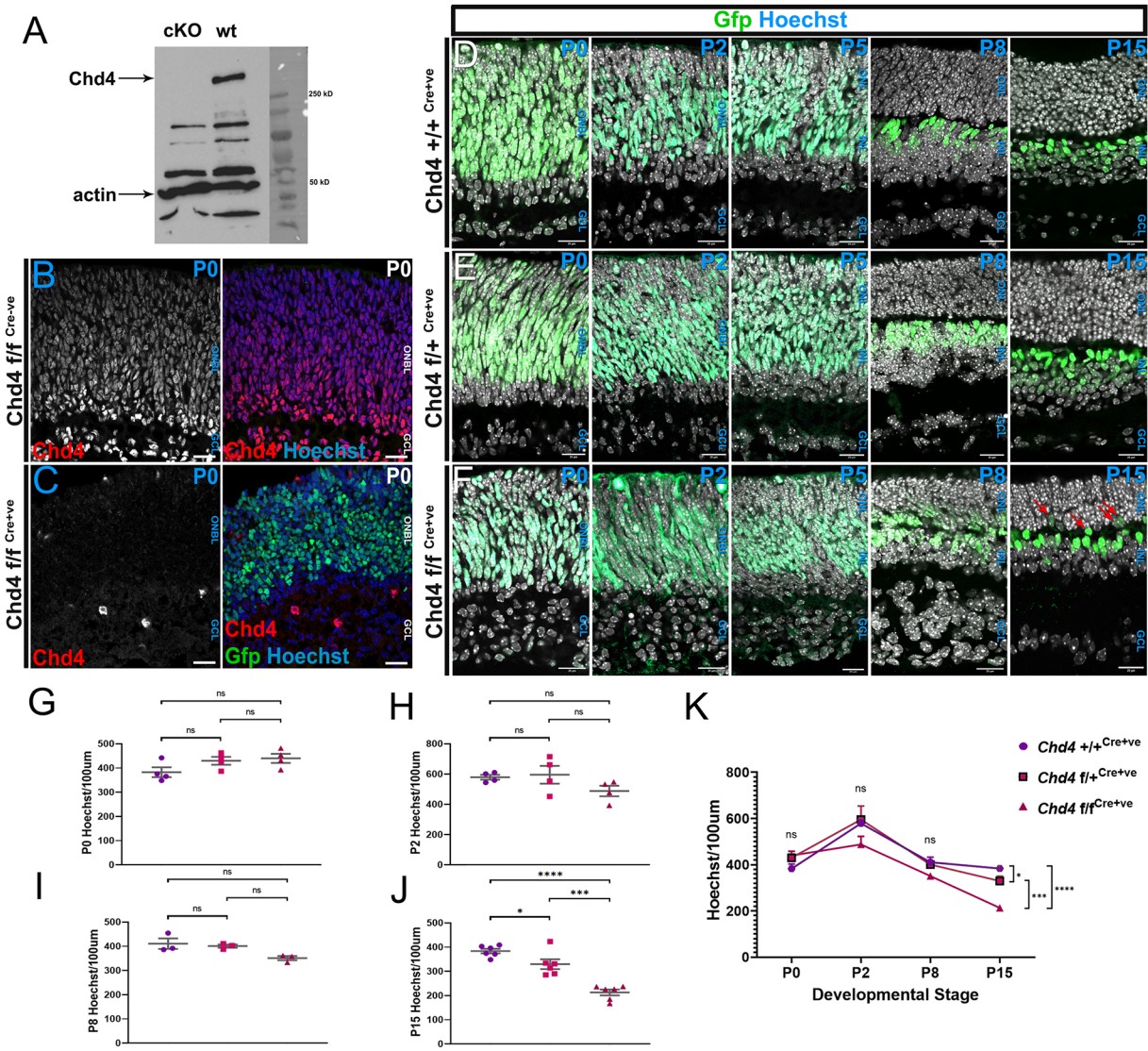

**Fig. 2. Chd4 is required for retinal histogenesis.** (A) Western analysis of Chd4 protein expression and actin as a control in *Chd4* cKO and wild-type retinas. (B,C) Chd4 immunostaining on P1 wild-type (B) or cKO (C) retinas. (D-F) Wild-type, chet and cKO retinas from P0, P2, P5, P8 and P15. Red arrows indicate ectopic GFP+ cells in the ONL. (G-K) Quantification of total Hoechst counts at P0, P2, P8 and P15, as indicated. All data are presented as mean ±s.e.m. *P<0.05, ***P=0.0001, ****P<0.0001; ns, not significant by one-way ANOVA with Tukey's multiple comparisons test. ONBL, outer neuroblastic layer; ONL, outer nuclear layer; INL, inner nuclear layer; GCL, ganglion cell layer. Scale bars: 20 μm.

corresponding to the wave of apoptosis previously shown to prune supernumerary bipolars and amacrines at ~P10 (Brzezinski et al., 2010; Dyer and Cepko, 2000; Katoh et al., 2010).

### Chd4 regulates retinal cell-type production

We next examined cell-type markers at P15, when development is complete. We found that cKO retinas had an almost twofold increase in the proportion of Rbpms+ RGCs as compared to controls (Fig. 3A,B). As a partial measure of amacrines, we counted Pax6+ cells within the INL. cKO retinas displayed a slight increase in INL amacrines when compared to wild type (Fig. 3C,D). These changes were observed in proportional counts, although not in absolute numbers (Fig. S5). For cones, we found that cKOs exhibited an

approximately twofold decrease in the percentage of cone arrestin+ cells when compared to wild-type and chet retinas (Fig. 3E,F).

During postnatal stages, RPCs generate amacrines, rods, bipolars and Müller glia. In the ONL, we counted marker-negative rods by excluding cone arrestin+ cones and GFP+ bipolars/Müllers. As expected from the thinning of the ONL (Fig. S4), rods were significantly reduced in *Chd4* cKO retinas as compared to wild type or chet (Fig. 3E,F). The proportion of strongly Otx2+ bipolar cells did not differ among the three genotypes at P15. However, cKOs exhibited ectopic Otx2+/GFP+ bipolar cells within the ONL (Fig. 3G,H). During retinal development, Müller glia are the latest-born cell type. Radially polarized Sox2+ cells were increased by almost twofold in the cKO, with many located within the ONL (Fig. 3I,J).

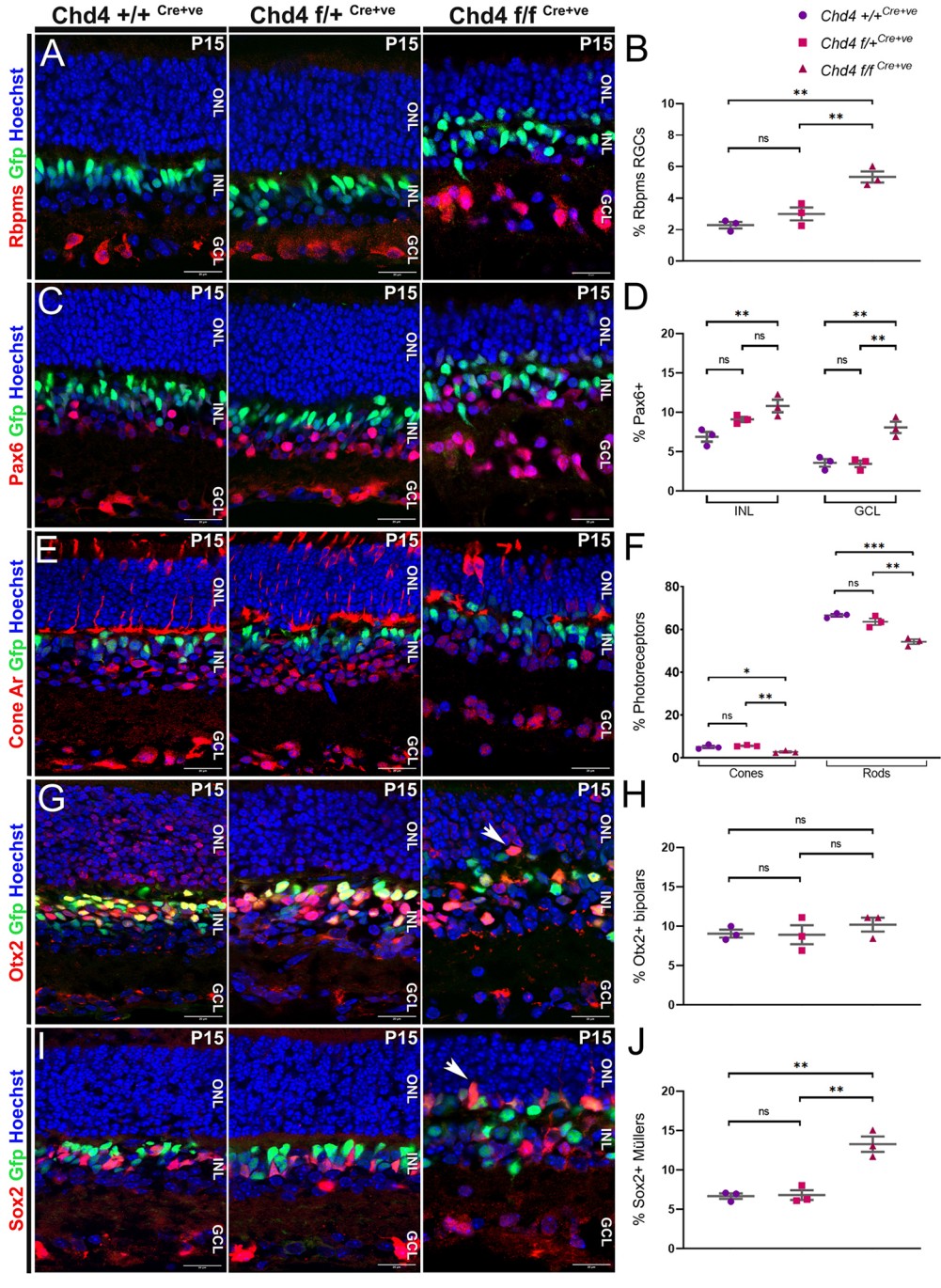

**Fig. 3. Shifts in retinal cell type composition in the *Chd4* cKO.** Marker staining was used to quantitate cell type proportions among the three genotypes at P15. (A,B) Rbpms+ RGCs. (C,D) Pax6+ amacrines (INL) along with RGCs (GCL). (E,F) Cone arrestin+ cones. GFP-negative and cone-arrestin negative ONL cells were counted as rods. (G,H) Brightly positive Otx2 cells were counted as bipolars. Arrow indicates ectopic bipolars in the ONL. (I,J) Radially polarized Sox2+ cells were counted as Müller glia. Data from A-J are also shown in Fig. S5. Arrow indicates ectopic glia in the ONL. All data are presented as mean±s.e.m. *P<0.05, **P<0.005, ***P<0.0005; ns, not significant by one-way ANOVA with Tukey's multiple comparison test. ONL, outer nuclear layer; INL, inner nuclear layer; GCL, ganglion cell layer. Scale bars: 20 μm.

### Chd4-dependent chromatin remodelling does not regulate progenitor proliferation

The observed shifts in cell type composition at P15 could potentially be explained by alterations in proliferation or cell death. For example, if self-renewing divisions were undermined, this might prematurely exhaust progenitors, leading to overproduction of early-born cells such as RGCs, and underproduction of late fates such as rods. We therefore examined retinas at earlier stages. At P0, when RPCs lose the competence to generate early-born cell types and rod production peaks (Carter-Dawson and LaVail, 1979; Young, 1985), Brn3a staining confirmed that RGCs were increased approximately twofold in the *Chd4* cKO (Fig. 4A-D) – similar to the increase observed at P15. The expansion in early-born neurons was also illustrated via the

RGC marker Rbpms (Fig. S6). Horizontal cells are the rarest early-born cell type, representing ~0.5% of the total retinal cell count (Jeon et al., 1998). We visualized horizontals by performing Lhx1 and calbindin co-staining, but did not observe a significant difference in horizontal cell numbers (Fig. S6E-I). Finally, we noted that the size of the progenitor pool – as reflected by the expression of the *Chx10-Cre-GFP* transgene – was not altered (Fig. 4A-E).

We next hypothesized that RGCs might be produced beyond their normal birth window. Postnatally, only trace levels of RGC production are observed in wild-type mice (Young, 1985). We therefore injected EdU at P0. After 2 days, we observed small numbers of newly born EdU$^+$/Brn3b$^+$ RGCs that were often localized apically, suggesting that they were migrating towards the

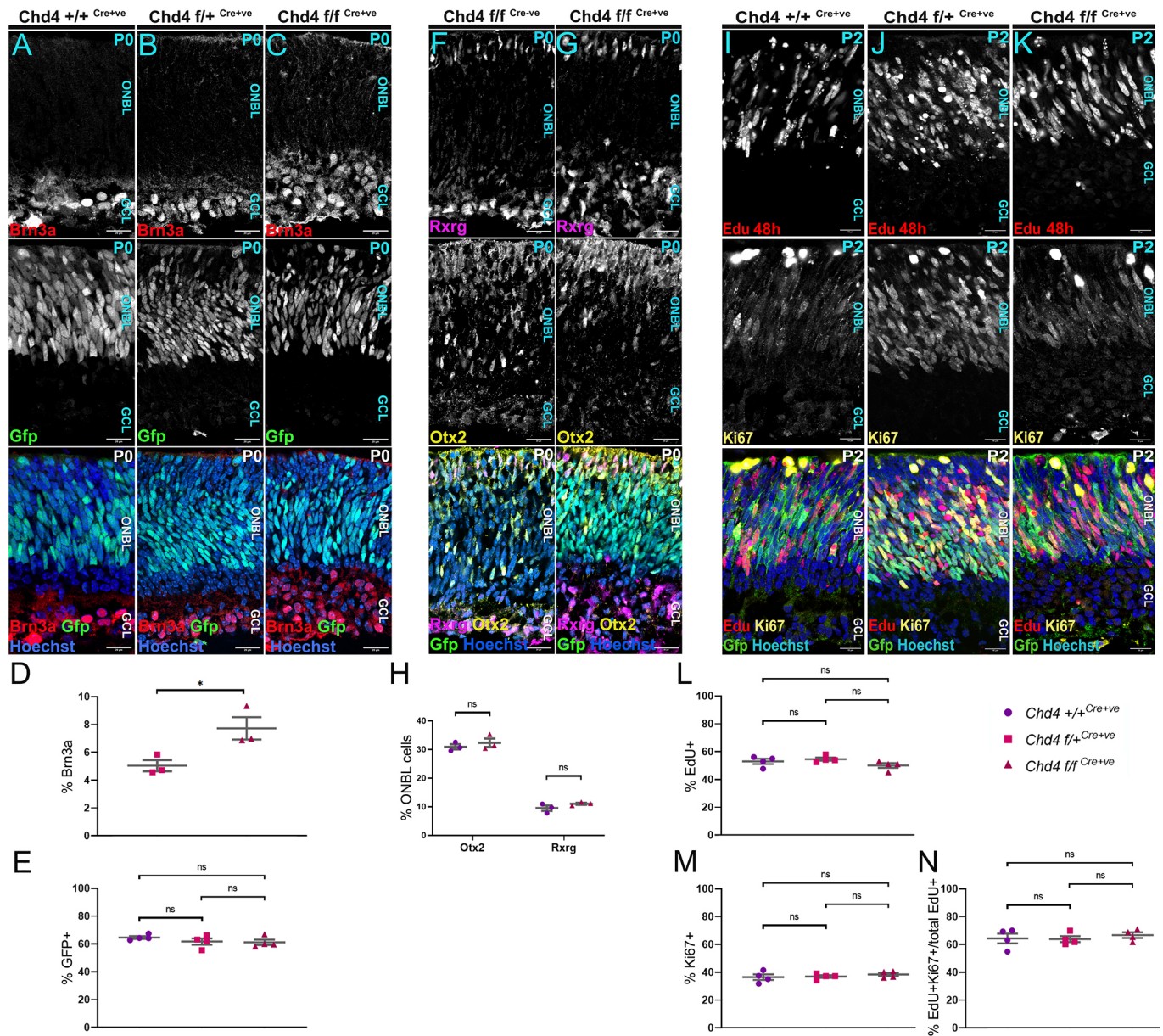

**Fig. 4. Fate shifts are independent of perinatal alterations in RPC proliferation.** (A-C) P0 retinal sections co-stained for the RGC marker Brn3a and imaged for GFP, which marks RPCs. (D,E) Percentage of Brn3a$^+$ cells (D) or GFP$^+$ cells (E). (F,G) P0 retinal sections were co-stained for the cone marker Rxrg and the photoreceptor precursor marker Otx2. (H) Percentage of Otx2$^+$ and Rxrg$^+$ cells between control and mutant retinas. (I-K) EdU was injected at P0. At P2, retinas were harvested and stained for EdU and Ki67. (L-N) Percentage of cells that were EdU$^+$ (L), Ki67$^+$ (M) or EdU$^+$Ki67$^+$ (N) as a percentage of total EdU$^+$ cells. Data from D,E are also shown in Fig. S5. All data are presented as mean±s.e.m. *$P<0.05$; ns, not significant by one-way ANOVA with Tukey's multiple comparisons test. ONBL, outer neuroblastic layer; GCL, ganglion cell layer. Scale bars: 20 µm.

GCL. However, very few of these cells were observed, and there was no marked difference between controls and cKOs (Fig. S7A-D). Similarly, EdU⁺ RGCs were not observed when EdU was injected at P1 and retinas were harvested at P8 (Fig. S7E). These results agree with the marked expansion of the GCL already observed by E16.5 and P0 in the *Chd4* cKO (Figs 2 and 4, Fig. S3), as well as scRNA-seq data (see below), which together indicate that supernumerary RGCs are produced during embryogenesis within their normal birth window.

Next, we hypothesized that RGCs might increase at the expense of the rods and cones that are generated during embryonic stages. We therefore counted Otx2⁺ photoreceptor precursors at P0, as well as Rxrg⁺ cones. Surprisingly, we found that proportions of early-born photoreceptors in *Chd4* cKO retinas were comparable to controls (Fig. 4F-H), suggesting that the decrease in cone arrestin⁺ cells observed at P15 might reflect subsequent defects in cone differentiation or survival. Thus, while RGCs expand and photoreceptors contract in number, these changes appeared not to be linked to a common fate decision.

Since rod photoreceptor production normally peaks between P0 and P2 (Carter-Dawson and LaVail, 1979; Young, 1985), we reasoned that RPCs might exhibit defects in proliferation that could undermine rod production at perinatal stages. We first examined phospho-histone H3⁺ mitotic cells, but found no difference between the genotypes (Fig. S2A-D). Next, we injected EdU to mark S-phase cells at P0. At P2, RPCs were co-stained for EdU and Ki67, which marks proliferating cells. Both EdU and Ki67 were comparable between the three genotypes (Fig. 4I-M). Double-labelled EdU⁺/Ki67⁺ RPCs that had undergone self-renewal were also not significantly different (Fig. 4I-N). These data indicate that *Chd4* does not affect proliferation dynamics at perinatal stages, in agreement with lack of significant differences in overall cell numbers observed up to P8. However, previous work in the developing neocortex had shown that loss of Chd4 led to elevated cell death (Nitarska et al., 2016). We therefore performed the TUNEL assay at P0, but changes in apoptotic frequency were not yet observed in the *Chd4* cKO versus controls (Fig. S2). Thus, while *Chd4* cKOs exhibited distorted cell-type proportions that could have arisen as a byproduct of premature RPC exhaustion or cell death, such effects were not yet evident perinatally during the normal peak of rod production.

### Chd4 regulates neurogenic competence at late stages

By P15, *Chd4* cKOs exhibit reductions in rods and a proportional expansion in Sox2⁺ glial cells, but also marked cell loss. To help determine whether these changes arose due to altered cell type production or through apoptosis, we next examined retinal development between P5 and P8. At P5, overall cell counts did not differ between controls (wild type/chet) and cKOs (Fig. 5A-D). However, rod photoreceptors expressing Otx2 and Nr2e3 were significantly reduced – both in absolute and proportional terms (Fig. 5A-E, Fig. S5). There was an overt expansion in Sox2⁺/GFP⁺ RPCs in *Chd4* cKOs (Fig. 5E). However, when we stained P5 retinas for activated caspase 3, we observed a significant elevation in apoptotic cells (Fig. 5F-H). Thus, the reduction in rod photoreceptors was concomitant with increased cell death in the *Chd4* cKO.

At P8, the laminar distribution of cells was markedly altered. In control retinas, almost all GFP⁺ cells were basal to the forming plexiform layer that divides the ONL and INL. However, in the *Chd4* cKO, many GFP⁺ cells were located within the ONL – apical to the plexiform layer (Fig. 5I-Q), which was more discontinuous in comparison to controls. In the ONL, GFP-negative photoreceptor precursors were also visibly reduced in the cKO, again suggesting

that late-stage rod production was reduced. To examine this more directly, we marked newly born cells via injection EdU at P1 and visualized their subsequent fates at P15. We found that the proportion of EdU⁺ cells was reduced in the ONL and increased in the INL (Fig. S8).

Next, we examined additional cell types. Tfap2a⁺ amacrines were not significantly altered, but Otx2⁺ bipolars were modestly but significant increased (Fig. 5I-K). Next, we examined Sox2. In the basal INL and GCL, Sox2 marks early-born cholinergic amacrine cells with large circular nuclei. These amacrines were significantly increased in the cKO (Fig. 5L-O; Fig. S9). More apically, radially polarized GFP⁺/Sox2⁺ cells were also significantly increased (Fig. 5L-O; Fig. S9). Since Sox2 marks both RPCs and Müllers, we next stained for markers that can distinguish between these cell types. Strikingly, while the proliferation marker Ki67 was restricted to the peripheral margins of the retina in controls, proliferating RPCs persisted throughout the retina in *Chd4* cKOs (Fig. 5P-R). Cell counts revealed a significant increase in RPC numbers (Fig. 5R). Next, we birthdated cell-type production from persisting RPCs in the *Chd4* cKO by injecting EdU at P8 (Fig. S10). At P15, control retinas exhibited EdU⁺ cells only at the peripheral margins of the retina, and EdU⁺ cells were often rod photoreceptors located within the ONL. By contrast, EdU⁺ cells were found throughout the central retina in the cKO, and while some ONL cells were labelled, the vast majority of EdU⁺ cells were Sox2⁺/GFP⁺ glia.

To definitively mark differentiated Müller glia, we next stained for Rlbp1. In controls, Rlbp1 expression stained brightly GFP⁺ cells in the basal INL. Strikingly, Rlbp1 was virtually absent from the *Chd4* cKO retina (Fig. 5S,T). Rlbp1⁺ cells could be seen only in patches of Cre mosaicism (Fig. S11). In the cKO, Rlbp1 was little expressed at P6 and P8, but was expressed by P15 (Fig. S11). Taken together, these data demonstrate that RPCs fail to differentiate into quiescent glia on schedule and accumulate in the *Chd4* cKO, but eventually differentiate.

### Loss of Chd4 results in divergent transcriptomic profiles

To understand how *Chd4* mutation shifts the transcriptional state of RPCs, we performed scRNA-seq at P1, prior to the marked distortions in proliferation, cell death and differentiation observed at later stages. To avoid batch effects, we used the Multi-seq barcoding approach (McGinnis et al., 2019), allowing us to compare biological replicates for three *Chd4* cKOs versus three littermate control retinas processed together within the same 10X Genomics Chromium well. Cells were sequenced to a depth of 23,190 reads and 1980 genes per cell for an estimated sequencing saturation of 49.6%. After demultiplexing and removing low-quality cells and doublets, our dataset retained 9776 cells, with 2152 control cells and 7624 cKO cells. Next, cell types were annotated using scDeepSort (Shao et al., 2021) to perform unsupervised label transfer based on a previously published retinal scRNA-seq atlas (Clark et al., 2019).

Next, we visualized the data using uniform manifold approximation and projection (UMAP; Fig. 6A). We noted that cells annotated as 'late RPCs' formed a wheel-like structure, from which a neurogenic 'stem' emerged, followed by a bifurcation towards amacrine cells, or alternatively towards photoreceptor precursors. Marker gene expression confirmed the fidelity of the cell-type annotation (Fig. S12). Next, we examined each replicate (pup) individually (Fig. 6B). To confirm the genotype of each barcode, we examined how *Chd4* and its paralogs were expressed in each replicate (Fig. S13). Focusing on RPCs, we observed that *Chd4* cKO cells exhibited a significant reduction in *Chd4* transcription as compared to controls, but that *Chd4* was not eliminated (Fig. 6C). However, this was

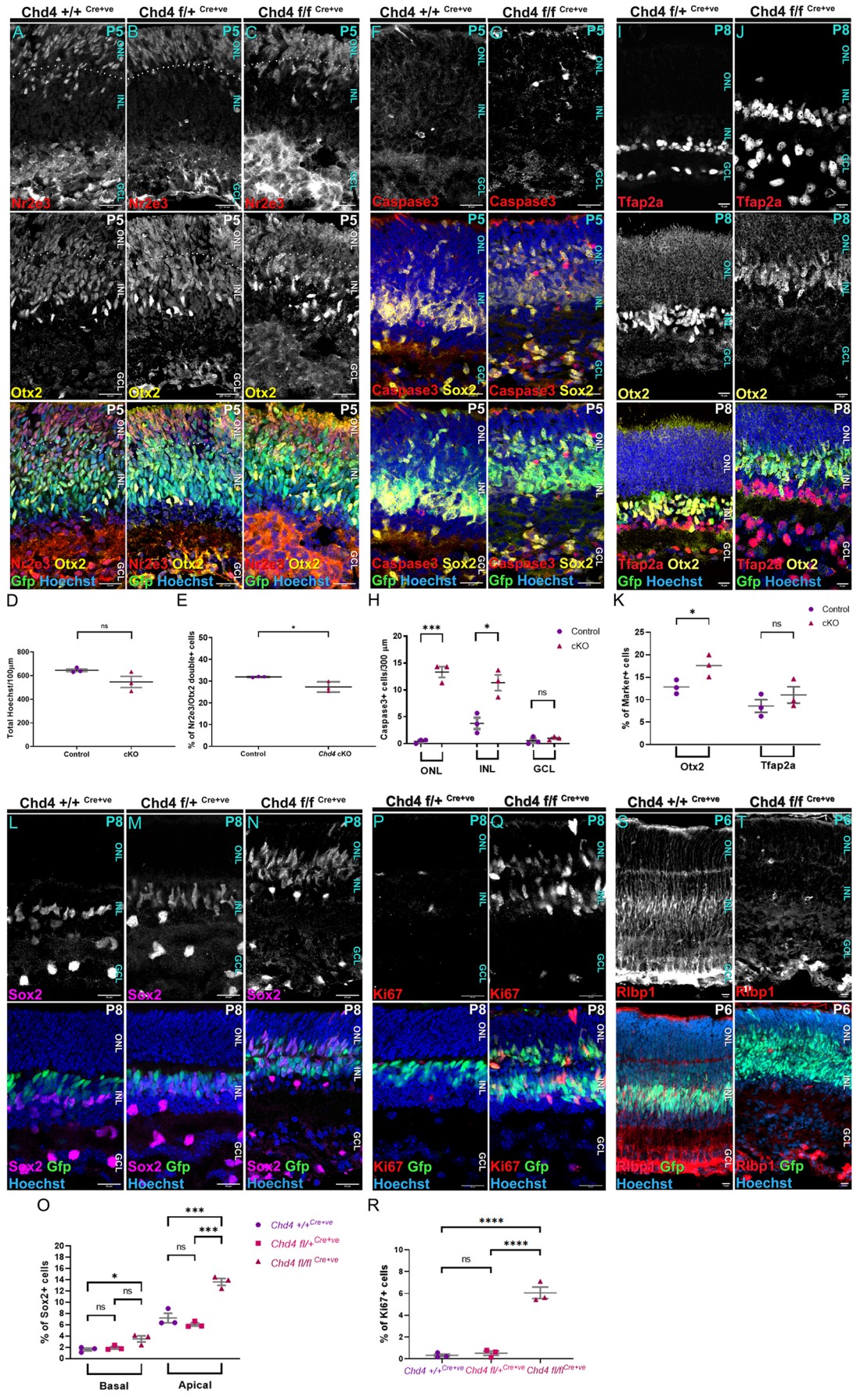

**Fig. 5.** See next page for legend.

**Fig. 5. Chd4 is required to terminate the retinal lineage.** (A-H) Marker staining was used to quantitate cell type proportions at P5. (A-C) Nr2e3+ rods and Otx2+ photoreceptor precursors. (D) Hoechst counts per 100 µm. (E) Nr2e3+/Otx2+ rod precursors. (F,G) Sox2+ RPCs and apoptotic cells, marked by active caspase 3. (H) Active caspase 3+ cells per 300 µm. (I-R) Marker staining was used to quantitate cell type proportions at P8. (I,J) Tfap2a+ amacrine cell Otx2+ bipolars. (K) Percentages of marker-positive cells, as indicated. (L-N) Cells brightly expressing Sox2 are cholinergic amacrines, whereas dimly positive, radially polarized cells are glia. (O) Quantitation of basal Sox2+ amacrines versus apical Sox2+ glia. (P,Q) Proliferative cells expressing Ki67 persist in the central retina in the cKO. Data from E and O are also shown in Fig. S5. (R) Quantitation of percentages of Ki67+ cells in the central retina. (S,T) Rlbp1 specifically marks Müller glia. All data are mean±s.e.m. *P<0.05, ***P=0.0001, ****P<0.0001. ns, not significant by two-tailed unpaired Student's t-test or one-way ANOVA with Tukey's multiple comparison test. ONL, outer nuclear layer; INL, inner nuclear layer; GCL, ganglion cell layer. Scale bars: 20 µm.

expected, since the loxP-flanked cassette does not excise the 3′ end of the gene (Williams et al., 2004). Since *Chd4* cKO has been shown to lead to a compensatory upregulation in its paralogs *Chd3* and *Chd5* (Clémot-Dupont et al., 2025; Yamada et al., 2014), we additionally examined these transcripts and found that both were significantly upregulated in *Chd4* cKO samples, as expected (Fig. 6C). Immunohistochemistry confirmed that Chd3 protein upregulated in *Chd4* cKO RPCs, further validating these observations (Fig. S14).

Next, we visualized *Chd4* cKOs versus controls. Control cells from each replicate clustered together in UMAP space (Fig. 5D). *Chd4* cKO cells overlapped with control cells, but were additionally shifted into novel parallel clusters that did not contain control cells (Fig. 6E). We found that significantly upregulated genes – including *Chd5* (Fig. 6F) and *Tcfl5* (Fig. 6G) – were expressed only in the novel clusters that appeared in *Chd4* cKO samples, but not in clusters occupied by control cells. These data likely indicate that our *Chd4* cKO samples exhibit marked alterations in gene expression across the full developmental trajectory, and that these changes are observed despite some probable mosaicism in cKO replicates (Fig. S13), as well as compensation from *Chd3* and *Chd5* paralogs.

We next identified differentially expressed genes (DEGs; Table S3). Focusing specifically on RPCs, *Chd4* cKOs exhibited both downregulated and upregulated DEGs (Fig. 6H). Downregulated DEGs included the transcription factor *Irx5*, which is involved in the specification of bipolar cell subtypes (Cheng et al., 2005) and the proneural gene *Ascl1*, which is necessary for rod and bipolar cell production (Tomita et al., 1996) (Fig. 6I). In accordance with the later expansion of the RPC pool, upregulated genes included *Apoe*, *Cdkn1a*, *Hes5*, *Mt1* and *Mt2*, which are all expressed in RPCs (Fig. 6H,I). Other upregulated genes included *Ifitm2*, *Tcfl5* and *Snhg11*, which were previously observed to upregulate in the *Chd4* cKO neocortex (Clémot-Dupont et al., 2025). We next called GO terms on these upregulated DEGs. Top terms included 'Growth', 'Cell population proliferation' and 'Cell death' (Fig. 6J), which are phenotypes that emerge in the cKO at later stages but were not yet evident in perinatal counts.

Despite the lack of obvious temporal shifts in cell-type production, we reasoned that a shift in the developmental stage of *Chd4* cKOs could be evaluated by comparing our dataset against other timepoints. We therefore integrated our dataset with an existing developmental scRNA-seq atlas (Clark et al., 2019). Using independent component analysis (ICA), we found that most of the timepoints in the published scRNA-seq atlas were arrayed in a logical continuum, with earliest embryonic samples at the origin, perinatal samples differing most along the first component (ICA1) and later postnatal samples differing most along the second component (ICA2; Fig. 6K). In

accordance with expectations, both P1 *Chd4* cKO and littermate control samples were localized near to P0 and P2 samples from the retinal atlas. However, we observed that the *Chd4* cKO samples were slightly shifted towards P5 samples, while littermate controls were located closer to P0 and P2 samples. To better visualize this potential shift, we plotted the integrated dataset in a comparison matrix (Fig. 6L). Both P1 control and cKO samples correlated most closely with P2 samples (control, 0.64; *Chd4* cKO, 0.52). Strikingly, *Chd4* cKOs correlated more strongly with P5 samples (0.47) than they did with E18.5 (0.27) or P0 (0.24) samples. By contrast, littermate control samples correlated more strongly with E18.5 (0.54) or P0 (0.48) samples, and much less well versus P5 (0.23), as would be expected. Thus, while a shift in temporal identity could not easily be discerned with respect to UMAP trajectories, both global gene expression and DEG signatures suggest that *Chd4* cKOs may be slightly accelerated in their temporal state.

## Chd4 regulation of chromatin occupancy and accessibility in RPCs

Next, we wished to determine how Chd4 regulates the genome. To examine the genome occupancy of Chd4, we performed CUT&RUN-seq on P1 wild-type and cKO retinas using a validated Chd4 antibody (Clémot-Dupont et al., 2025; Yamada et al., 2014). We additionally examined Mbd3, which is specific to the NuRD complex. Visual comparison of these datasets revealed correspondence between Chd4 and Mbd3 (Fig. 7A). Across the genome, Chd4 occupied ~10,000 peaks in wild-type retinas, which was comparable to peak numbers observed in the neocortex and cerebellum (Clémot-Dupont et al., 2025; Yamada et al., 2014). Mbd3 occupied ~3500 peaks, with most of these peaks co-occupied by Chd4 (Fig. S15A-C). In *Chd4* cKO retinas, Chd4 and Mbd3 peak numbers were drastically reduced to ~2500 and 1000, respectively (Fig. S15A). When compared to published retinal ChIP-seq data (Aldiri et al., 2017), we found that approximately two-thirds of the Chd4 peaks localized to gene promoters marked by H3K4me3 (Fig. S15B).

To directly visualize the nucleosome remodelling activity of Chd4 in RPCs, we performed ATAC-seq on two wild-type and two cKO littermates, by sorting RPCs marked by the *Chx10-Cre-GFP* transgene at P1. Inspection of the loxP-flanked cassette revealed almost complete excision in the cKO, validating the sorting strategy (Fig. S16). To identify differentially accessible regions (DARs) in *Chd4* cKO RPCs, we next performed diffbind analysis (Fig. 7B), yielding approximately 10,000 DARs between control and mutant RPCs. Most DARs exhibited increased accessibility (Fig. 7A-C). While the NuRD complex has previously been shown to decommission some regulatory elements, we found that most DARs were still nominally accessible in the control datasets (Fig. 7C). More surprisingly, most of these DARs exhibited little Chd4/NuRD complex occupancy (Fig. 7B), indicating a probable 'kiss-and-run' transient interaction or indirect regulation. By contrast, DARs that were reduced in accessibility in *Chd4* cKOs appeared to be directly bound, suggesting that loss of the NuRD complex footprint might drive the effect.

To determine how differential accessibility might relate to gene expression, we performed peak-to-gene annotation. We selected only gene-proximal peaks (within 5 kb upstream and 1 kb downstream of the gene body inclusive) in order to filter the overall peak number. Gene proximal DARs were associated with ~1600 genes (Table S3). Gene ontology analysis showed that neuron fate commitment, neurogenesis and neuron differentiation were highly enriched (Fig. 7D), suggesting potential misregulation of these processes.

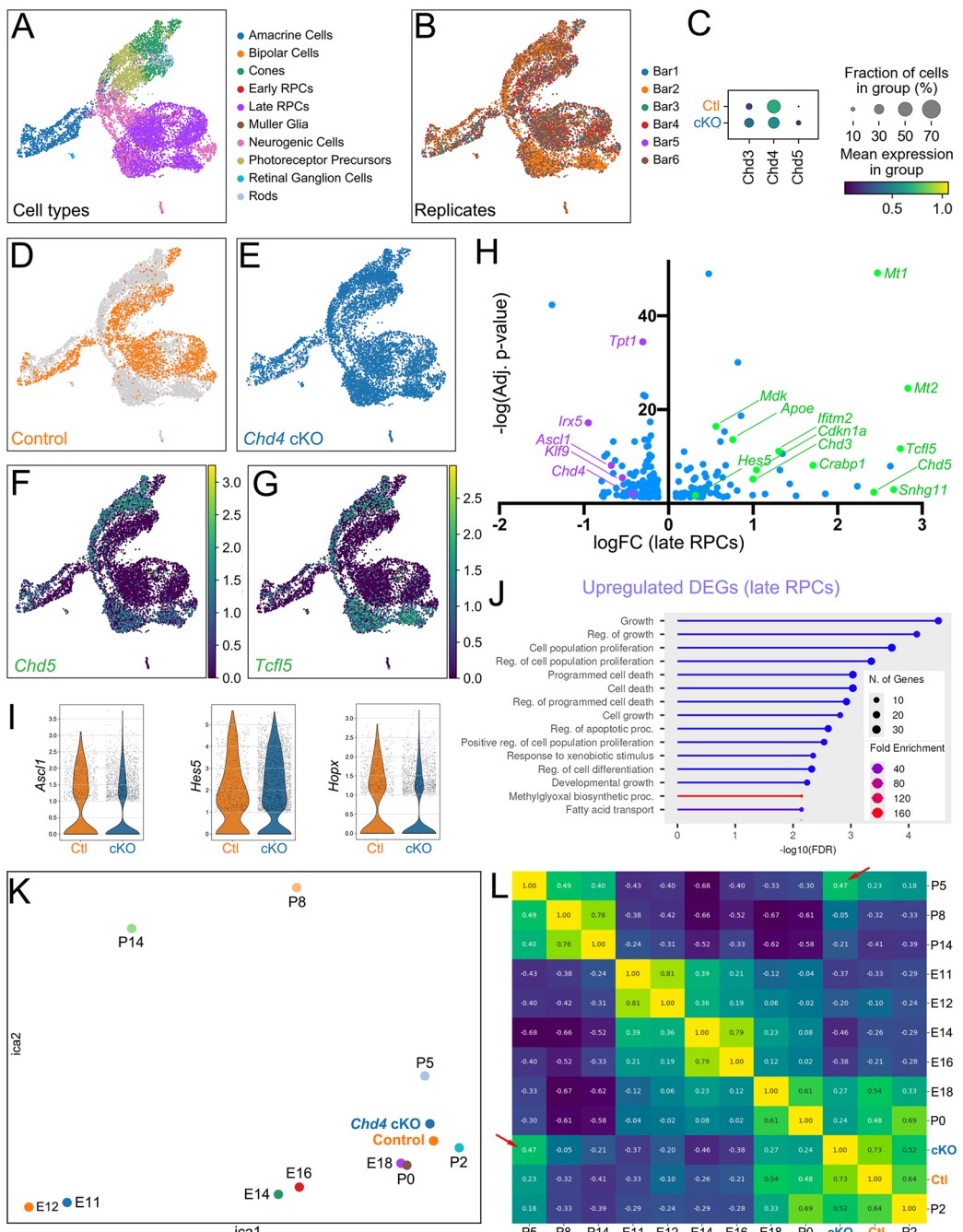

**Fig. 6. Loss of *Chd4* leads to global transcriptional dysregulation.** (A) Leiden UMAP clustering of 9776 single cells from P1 control (*n*=3) or cKO (*n*=3) littermates. Cell types were annotated via unsupervised label transfer from a published atlas of retinal development (Clark et al., 2019). (B) UMAP projection of each demultiplexed sample. (C) Dotplot comparing the expression of *Chd4* and its paralogs in late RPCs from control versus cKOs. (D,E) Comparison of control versus cKO cells. (F,G) UMAP projection of the expression of *Chd5* (F) and *Tcfl5* (G). (H) Volcano plot of differentially expressed genes (DEGs) from late RPCs in *Chd4* cKO samples versus control (adj. *P*-value<0.05; LogFC>0.4). (I) Violin plots comparing *Ascl1*, *Hes5* and *Hopx* expression in wild-type versus cKO late RPCs. (J) GO term analysis of significantly upregulated DEGs. (K) Independent component analysis (ICA) comparing wild-type and cKO scRNA-seq data with a published retinal RNA-seq atlas (Clark et al., 2019). (L) Pairwise comparison correlation matrix heatmap of the ICA analysis. Arrows indicate the elevated correlation between the cKO dataset and the P5 samples of the retinal atlas.

To determine how changes in accessibility affect gene expression, we examined genes associated with proximal DARs in our scRNA-seq dataset. We found that DAR-associated genes overlapped with only ~10% of DEGs. However, more extensive overlap was observed with Chd4 CUT&RUN-seq peaks, with more than half of the DEGs and approximately one-third of DAR-associated genes directly occupied by Chd4. Additionally, 33 target genes were common

between all three of the datasets (Fig. 7E; Table S3), including genes such as *Cited2* (Fig. 7A), *Mbnl2*, *Jund* and *Plagl1* (Fig. S17). In accordance with these observations, when we generated a gene scoring module for genes associated with a proximal DAR, we found a slight but significant upregulation in the *Chd4* cKO (Fig. 7F), suggesting that changes in accessibility correlate with transcription. We next measured accessibility across gene bodies using LIMMA,

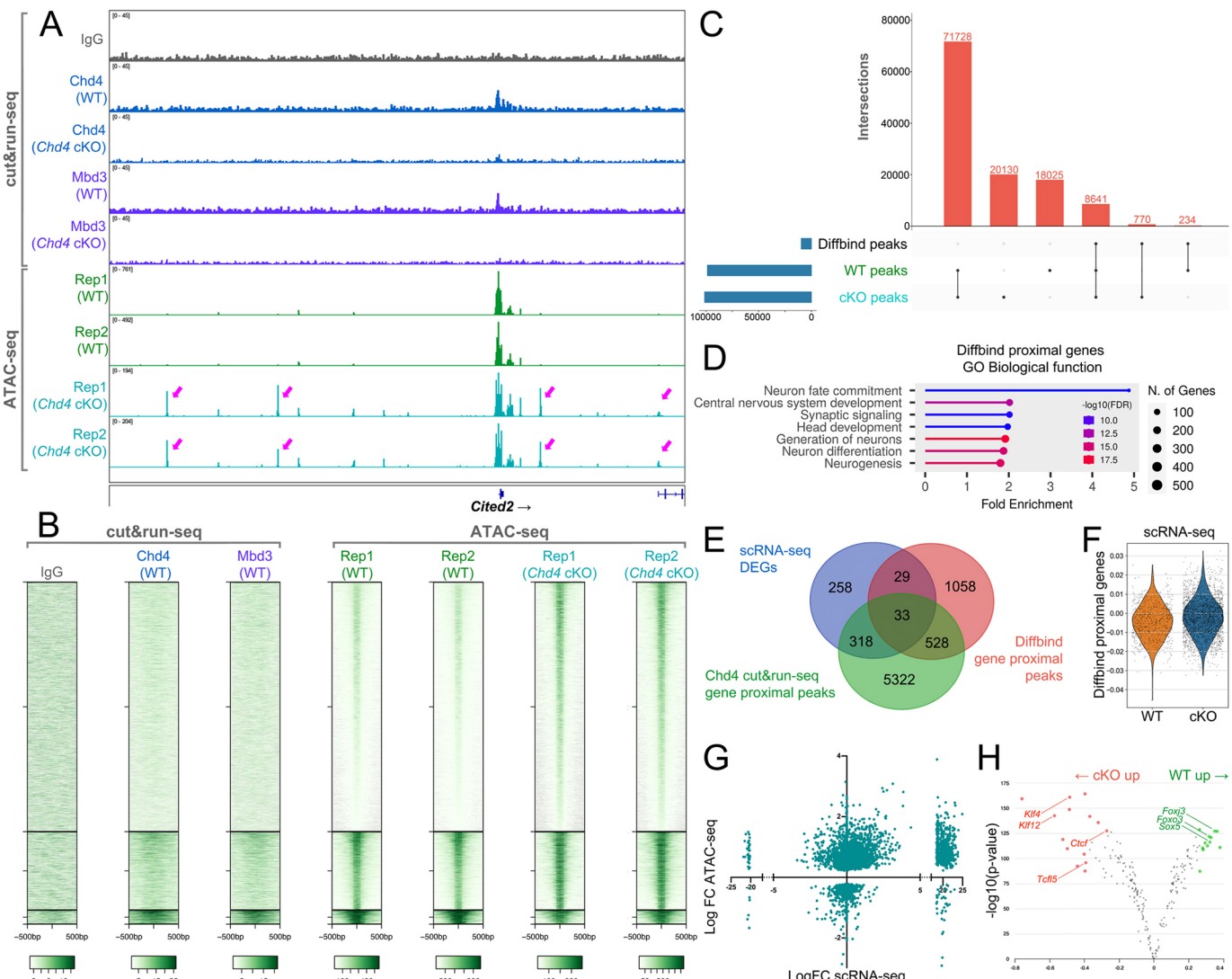

**Fig. 7. *Chd4* restricts chromatin accessibility in RPCs.** (A) CUT&RUN-seq and ATAC-seq tracks showing called peaks at *Cited2* locus from P1 control and *Chd4* cKO RPCs. Arrows indicate ectopic peaks present in the cKO samples but not in control. (B) CUT&RUN-seq and ATAC-seq datasets centred on differentially accessible regions (DARs) identified via Diffbind. (C) Upset plot comparing the overlap between ATAC-seq peaks comparing control and cKO datasets versus DARs. (D) GO terms analysis of Diffbind proximal genes. Diffbind proximal peaks were assigned to genes located within 5 kb upstream and 1 kb downstream of the peak. (E) Integration of DEGs from the scRNA-seq analysis with genes associated with gene-proximal DARs and Chd4 CUT&RUN-seq gene-proximal peaks. (F) Violin plot of scRNA-seq gene scores for genes associated with a proximal DAR, comparing control versus *Chd4* cKO RPCs. (G) Integration of scRNA-seq gene expression with gene body accessibility measured via LIMMA. (H) Footprinting analysis on accessible peaks using the TOBIAS algorithm.

which allowed us to integrate accessibility data for each gene body rather than on an individual peak-by-peak basis. Comparing fold-changes in gene accessibility versus fold-changes in transcription, we observed a significant ($P<0.0001$) positive correlation between increased accessibility and transcriptional upregulation across the genome (Fig. 7G).

Lastly, we performed footprinting analysis on accessible peaks in order to identify differential transcription factor occupancy using the TOBIAS algorithm (Fig. 7H) (Bentsen et al., 2020). Using this approach, we found that Ctcf was one of the over-represented motifs in *Chd4* cKO RPCs. This suggests that the loss of Chd4 might lead to increased recruitment of Ctcf to sites that are typically inaccessible, resulting in disorganization of genome looping, as previously shown in cerebellar granule cells (Goodman et al., 2020). Additionally, Tcfl5 motifs were also over-represented in mutant RPCs, corroborating increased *Tcfl5* transcript levels observed in

the scRNA-seq data. Taken together, these data suggest that Chd4 may have a broad role in restricting nucleosome accessibility and consequent transcription across the genome, which might stabilize the RPC identity and drive self-renewal.

## DISCUSSION

Although temporal transitions play a crucial role in diversifying neural progenitor lineages, the underlying molecular mechanisms are poorly understood. Here, we have addressed the role of nucleosome remodelling in regulating the chronology of cell type production from retinal progenitors. We hypothesized that Chd4-dependent chromatin remodelling would be required for the competence transition between embryonic and postnatal modes of neurogenesis. *Chd4* cKOs accordingly exhibited increases in early-born RGCs and amacrines, and later-born rods were drastically decreased (Fig. 8A,B), initially suggesting a prolongation of early

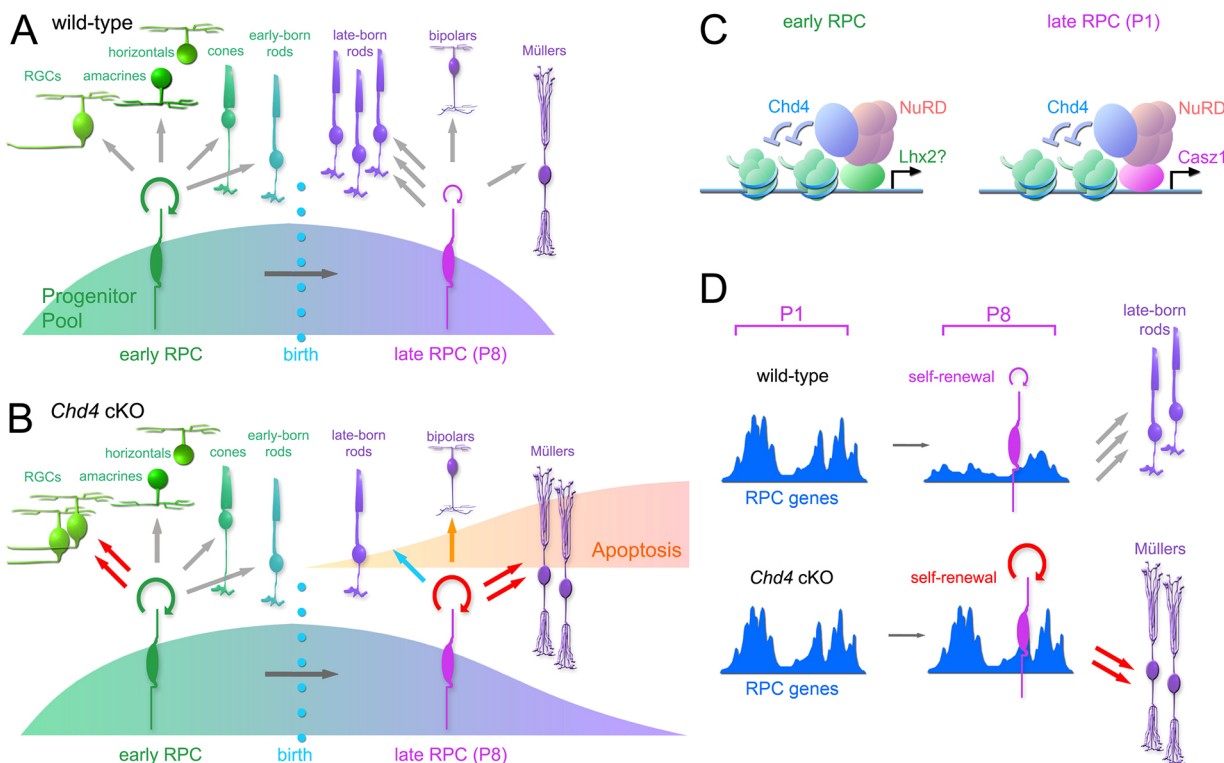

**Fig. 8. Requirement for Chd4 in retinal development and gene regulation.** (A,B) Model of lineage alterations in wild-type (A) versus *Chd4* cKO (B) retinas. Red arrows indicate that cell production increases; orange arrows indicate a slight increase in cell production; blue arrows indicate a decrease in cell production. (C) Chd4 may associate with different transcription factors at different developmental stages in order to control cell fate specification. (D) Model for the role of nucleosome remodelling in temporal progression of RPCs. In the absence of Chd4, expressed genes remain more accessible, reinforcing the progenitor state.

competence. However, this proved not to be the case. Indeed, scRNA-seq profiling suggested that *Chd4* cKOs were instead slightly accelerated in their temporal state. Moreover, we observed that cycling RPCs persisted beyond their normal developmental window and differentiated into glia much later than controls. Taken together, these data suggest that Chd4-dependent nucleosome remodelling regulates the temporal transition that terminates the retinal lineage, but does not control earlier competence transitions.

While distortions in cell type composition were observed prior to the significant cell loss that occurs between P8 and P15, alterations cell death may nonetheless be responsible for the observed shifts in cell type composition. Increases in early-born fates – including RGCs and cholinergic amacrines, occurred without a complementary decrease in alternative fates. While early hyperproliferation might explain the expansion of early-born fates, RGC apoptosis might instead be reduced. Many supernumerary RGCs die at perinatal stages, but there is an earlier wave of embryonic RGC cell death that we did not investigate (Farah, 2006; Pequignot et al., 2003). At birth, rod and cone numbers are not altered in the *Chd4* cKO, but they become significantly reduced by P5. These reductions corresponded with a significant elevation in apoptosis. While it is therefore difficult to exclude the notion that photoreceptors are selectively eliminated at late developmental stages, the concomitant expansion of cycling RPCs in the *Chd4* cKO suggests that, in addition to apoptotic loss, RPC self-renewal is increased, which we think would likely undermine rod production at late stages.

## Molecular control of progenitor competence

Competence determinants were first identified in *Drosophila* neuroblasts (Holguera and Desplan, 2018). Transcription factor cascades were shown to progressively modify progenitor potential, allowing neuroblasts to produce sequences of different neurons. In vertebrate lineages, epigenetic processes were the first regulators of temporal competence states to be identified (Hirabayashi et al., 2009; Takizawa et al., 2001), but transcription factor cascades that are analogous to *Drosophila* temporal factors were subsequently identified. It remains unclear whether transcription factors and heterochromatic determinants converge. However, multiple temporal transcription factors, including Ikzf1, Casz1 and Foxp1 have been shown to interact with the NuRD complex in various contexts (Chokas et al., 2010; Kehle et al., 1998; Kim et al., 1999; Liu et al., 2015; Mattar et al., 2021). We note that *Chd4* cKOs resemble *Casz1* cKOs, but have a stronger phenotype. Intriguingly, the *Drosophila* orthologue *castor* is similarly required to terminate neuroblast lineages (Maurange et al., 2008). By contrast, *Ikzf1* mutants underproduce early-born cell types (Elliott et al., 2008; Javed et al., 2023), in contrast to the *Chd4* cKO, suggesting that Ikzf1 likely acts in a NuRD-independent fashion. Since Ikzf1 regulates early competence, these observations imply that different temporal transitions are governed by different mechanisms – despite the fact that Chd4 is a known co-factor of Ikzf1. While *Ikzf1* and *Casz1* mutants have relatively mild phenotypes, *Chd4* abrogation leads to marked deficits in retinal lamination. Chd4 may be required for neurite outgrowth, as shown previously in cerebellar granule neurons (Yamada et al., 2014). The greater severity of the Chd4 phenotype suggests that, besides Casz1, additional transcription factors may interact with Chd4/NuRD in a stage-dependent fashion (Fig. 8C). Future work will be required in order to better understand the mechanisms that control retinal histogenesis.

In agreement with our results, distinctions between the epigenetic mechanisms that regulate different temporal transitions have been previously described in cortical lineages. Although polycomb was previously shown to regulate multiple competence transitions, including the transition between the production of early versus late-born neurons (Morimoto-Suzki et al., 2014; Zhang et al., 2023), Tsuboi et al. had already found that NuRD regulated the switch between neurogenesis and gliogenesis, but did not regulate earlier temporal transitions (Tsuboi et al., 2018). While Nitarska et al. previously showed that Chd4 is specifically required for the production of upper-layer cortical neurons (Nitarska et al., 2016), this phenotype appears to be driven in part by stage-specific defects in progenitor proliferation. Thus, even though Chd4 has striking effects on the production of early- versus late-born neurons in both the cortex and retina, we conclude that Chd4-dependent chromatin remodelling is not required for the transition between early versus late neurogenic competence.

In cortical lineages, heterochromatic chromatin remodelling complexes such as NuRD and polycomb were previously shown to regulate the switch from neurogenic to gliogenic competence (Hirabayashi et al., 2009; Tsuboi et al., 2018). However, in the retina, glial cells appear to be overproduced via a failure to exhaust cycling RPCs. Moreover, while the production of late-born rods was decreased, bipolar neurons were slightly increased at P8. Accelerated gliogenesis would presumably truncate the normal RPC lineage – leading to a balanced loss in rods and bipolars. Unlike cortical lineages, RPCs do not convert into true glioblasts. Instead, RPC lineages can terminate via Müller differentiation (Rulands et al., 2018), with RPCs perhaps directly converting into Müllers (Lyu et al., 2021; Shiau et al., 2021). The observation that cycling RPCs persist in the *Chd4* cKO retina at late stages suggests that RPCs become stalled within this mode of lineage termination. In this respect, *Chd4* cKOs resemble *Lhx2* cKOs (Gordon et al., 2013), which would be consistent with biochemical and functional data linking Lhx2 to NuRD in the cortex (Muralidharan et al., 2017). Intriguingly, RPCs in *Nfia/b/x* triple mutants were previously shown to persist beyond the normal termination of neurogenesis, generating supernumerary rods, and underproducing bipolars and Müllers (Clark et al., 2019). The complementarity of the *Chd4* and *Nfia/b/x* cKO phenotypes might perhaps illustrate two alternative mechanisms that act in parallel to terminate the retinal lineage.

### The role of nucleosome remodelling in retinal progenitors
To understand how Chd4 regulates the genome, we focused on perinatal stages, where birthdating analyses showed that RPCs have altered cell-type production, but did not yet exhibit alterations in proliferation or apoptosis. Our scRNA-seq results showed that loss of *Chd4* led to widespread transcriptomic dysregulation. Downregulated genes included the late proneural gene *Ascl1*, whereas a number of RPC genes were upregulated, likely presaging the later requirement for *Chd4* to promote the exhaustion of RPC proliferation at the end of retinal development. Upregulated DEGs were also linked to proliferation and apoptosis, which were indeed increased at later stages.

To directly visualize the nucleosome remodelling activity of Chd4, we performed ATAC-seq on sorted RPCs, revealing that, in the absence of *Chd4*, there were modest increases in accessibility at thousands of regulatory elements, which correlated with increased transcription at associated genes. Similar functions have previously been described in cerebellar granule neurons, where Chd4 depletion led to a widespread increase in genome accessibility (Goodman et al., 2020; Yamada et al., 2014). These modest but genome-wide

changes in accessibility have also been reported in embryonic stem cells, where NuRD was required to suppress inappropriate gene expression and for the efficient activation of genes required for differentiation (Bornelov et al., 2018; Montibus et al., 2024). Taken together, our data suggest that, without Chd4, the downregulation of RPC gene expression signatures is undermined, leading to a reinforcement of the RPC state (Fig. 8D).

In the developing retina, Chd4 directly occupied ~10,000 peaks, similar to what was reported in the cerebellum and neocortex (Clémot-Dupont et al., 2025; Yamada et al., 2014). However, Mbd3 CUT&RUN-seq resulted in only 3500 peaks, perhaps suggesting NuRD independent functions of Chd4. Recent studies have shown that, apart from the NuRD complex, Chd4 can also interact with the transcription factor Adnp in order to form the ChAHP complex, which in turn can regulate cortical neurogenesis (Clémot-Dupont et al., 2025; Ostapcuk et al., 2018). The Chd4 phenotype might also arise due to the dysregulation of genome looping. Ctcf was one of the over-represented footprints in *Chd4* cKO RPCs. Chd4 has previously been shown to regulate Ctcf binding and thereby genome architecture (Goodman et al., 2020). Chd4 might also regulate cohesin through the ChAHP complex, since Adnp and Ctcf have been shown to bind the same DNA motif (Kaaij et al., 2019). Future studies will be required to determine whether Chd4 regulates neurogenic competence through NuRD, ChAHP and/or via interactions with additional co-factors, such as temporal transcription factors.

## MATERIALS AND METHODS
### Animals
Mouse work was conducted according to the guidelines laid out by the Canadian Council of Animal Care under the supervision of the uOttawa Animal Care and Veterinary Service (ethical protocols OHRI-2856, OHRI-2867, OHRI-3949 and OHRI-4029). *Chd4* floxed alleles (*Chd4*$^{f/f}$) were graciously donated by the Katia Georgopoulos laboratory (Williams et al., 2004) and backcrossed onto the C57BL/6J background (obtained from Jackson Laboratories; RRID: IMSR_JAX:000664). *Chx10-Cre-GFP* transgenic mice (RRID:IMSR_JAX:005105) (Rowan and Cepko, 2004) were generously provided by the Catherine Tsilfidis laboratory. Both males and females were used for all experiments. Genotyping primers are presented in Table S1.

### Histology
For embryonic stages, whole heads were collected and fixed in 4% PFA overnight at 4°C. This was followed by three washes in 1×PBS for 5 min each followed by immersion in 20% sucrose in PBS at 4°C overnight. The next day they were subjected to three washes in 1×PBS for 5 min each and submerged in 1:1 solution of OCT:20% sucrose in 1×PBS overnight at 4°C. Subsequently, heads were embedded in the OCT compound and were stored at −80°C. For P0 and P2, a small slit was introduced between the lens and choroid to allow fixative to penetrate the eye. Eyes were then fixed in 4% PFA for 15 min. For P8 and P15, lenses were removed to generate eye-cups. This was followed by 4% PFA fixation for 2-3 min. After fixation, tissues underwent three washes in 1×PBS, and were transferred to 20% sucrose in PBS at 4°C for 2-3 h. They were finally immersed in the OCT compound for storage at −80°C. For whole-mount staining, fixed retinas were directly immersed in staining solutions, as described below for sections.

Coronal cryosections (14 μm) were collected onto Superfrost Plus slides (Fisher) and processed for immunofluorescence as described previously (Mattar et al., 2015, 2021). Briefly, the tissue sections were washed thrice in 1×PBS followed by antibody incubation at 4°C overnight. For whole-mount staining, retinas were immersed in antibody solutions at 4°C for 2-4 days. Primary antibodies were diluted to 1:100 in blocking buffer (PBS supplemented with 0.4% Triton X-100, 3% w/v BSA and 1:5000 Hoechst 33342). After washing with 1×PBS, Alexa 555- or 647-conjugated secondary antibodies were added at a dilution of 1:1000 in blocking

buffer and incubated for 2-3 h at room temperature. Finally, the coverslips were mounted using Mowiol mounting media [12% w/v Mowiol 4-88, 30% w/v glycerol, 120 mM Tris-Cl (pH 8.5), 2.5% DABCO] and stored at 4°C until imaged. Antibody information is presented in Table S2.

### Imaging and cell counting
Images were acquired on an LSM900 confocal microscope (Zeiss), using a 63X objective (Plan-Apochromat 63×/1.40 Oil DIC f/ELYRA) with a 0.5X digital zoom. We avoided retinal fields that exhibited significant *Chx10-Cre-GFP* mosaicism as determined by GFP epifluorescence. We found that mosaicism in progenitors could be directly visualized up to P8, but could not be determined at later stages when GFP became expressed in bipolar cells. We therefore imaged retinal fields at random in P15 experiments. The images were tiled using Zen software (Zeiss) where required. For each biological replicate, four optical sections (single Z planes) of the peripheral retina were used to generate the cell counts. Manual cell counting was performed using Fiji (ImageJ), wherein each optical section analysed was cropped to a constant width of 100 μm. Images were further processed using Adobe Photoshop CS3 (Adobe) software.

### EdU incorporation
P0 and P1 pups were injected intraperitoneally with 10 mM of EdU (Invitrogen, C10640) and eyes were harvested at P2 or P15 as indicated in the figure legends. EdU staining was performed on retinal sections using the Click-iT Plus EdU Cell Proliferation Kit for Imaging (Invitrogen, C10640) according to the manufacturer's protocol. In the instances where EdU was co-stained with other antibodies, the tissues were first processed for immunostaining with primary antibodies before the Click-iT staining reaction.

### Statistical analysis
Statistical analysis for image count data was performed using Microsoft Excel and GraphPad Prism version 8 (GraphPad) software. *N* values refer to biological replicates and each data point denotes a single biological replicate. For cell counting, a minimum of three biological replicates was quantified, and statistical analysis was performed via one-way ANOVA with Tukey's multiple comparison test or two-tailed unpaired *t*-tests, as appropriate. We did not perform statistical analyses to predetermine sample sizes. The count data are presented as mean±standard error of mean (s.e.m.). *$P<0.05$; **$P<0.05$; ***$P<0.005$; ****$P<0.0005$. ns, not significant.

### Western blot
Western blotting was performed as previously described (Mattar et al., 2021) with some modifications. P0 dissected retinas were homogenized in RIPA lysis buffer with protease inhibitors (cOmplete, Mini, EDTA-free; 11836170001; Millipore Sigma) and incubated on ice for 15 min. Thereafter, they were sonicated on ice using a Cole-Parmer Ultrasonic Homogenizer (RK-04711-45) at 20% amplitude with an 8 s pulse followed by a 30 s interval for a total of three pulses and centrifuged at 21,000 *g* at 4°C for 15 mins. The supernatant was collected, and the protein concentration was quantified using BCA assay. Approximately 40 μg of protein lysate was separated on a 6-10% gradient SDS-PAGE gels and semi-dry transferred onto PVDF membranes (Millipore).

### Retinal dissociation
Retinas were dissociated using papain [0.003 N NaOH, 100 U papain solution (Worthington), 0.4% DNaseI (Millipore Sigma, 04716728001) and 2 mg L-cysteine crystal in 1×PBS] for 8 min at 37°C. Subsequently, papain buffer was aspirated and replaced with LO-OVO solution [1×LO-OVO (Bio Basic) and 0.4% DNaseI in 1×PBS]. Thereafter, the retinal tissue was triturated and centrifuged for 11 min at 200 *g* at room temperature. The retinal cells were then re-suspended in 1×PBS.

### ATAC-seq
Two biological replicates were used for each genotype: control (wild type) and mutants (cKO). P1 retinas from each biological replicate were dissected to make a dissociated suspension as described above. Subsequently, 75,000 GFP⁺ cells were flow-sorted and used for the ATAC-seq assay as described

previously (Buenrostro et al., 2015; Herrera et al., 2023) using the Nextera library kit (Illumina).

The analysis of the ATAC-seq dataset was performed using the Galaxy interface (Galaxy, 2024). After the initial quality control, the NGS adapters were trimmed using Trimmomatic, and the reads were mapped to the mm9 reference genome using Bowtie2. After merging the alignment files via Samtools merge, peak calling was performed with MACS2. Differential peak detection and analysis were performed using DiffBind and Limma. DiffBind peaks were sorted via k-means clustering with Seqplots and annotated to nearby genes using GREAT. ATAC-seq and CUT&RUN-seq data have been deposited in GEO under accession GSE266039.

### Multi-seq
Multi-seq was performed as described previously (Herrera et al., 2023; McGinnis et al., 2019) with some modifications. For this assay, we utilized three biological replicates each for control (chet) and cKO samples, for a total of six replicates. P1 retinas from each biological replicate were dissociated. Approximately 250,000 dissociated cells per replicate were then barcoded by incubating with 'anchor' and 'co-anchor' lipid-modified oligonucleotides graciously provided by the Gartner lab (University of California San Francisco, USA). Barcode oligonucleotides were purchased from Integrated DNA Technologies (see Table S1). Barcodes 1, 3 and 4 were used to tag individual control replicates, while barcodes 2, 5 and 6 tagged individual mutant replicates. The respective barcode sequences are shown in Table S1. Individual replicates were co-incubated with barcode oligonucleotides and pooled into a single tube at a 1:1 ratio. Approximately 20,000 pooled cells were used in a single Chromium run (3′ Library & Gel Bead Kit v2, PN-120237, 10X Genomics).

The resulting expression library FASTQs were processed using CellRanger (10X Genomics). The deMULTIplex workflow was used to perform quality control to remove doublets along with cells that lacked the barcodes. Output files were filtered and analysed using Scanpy version 1.9.190. Genes detected in fewer than three cells were removed from the analysis. Low-quality cells (less than 5000 genes detected, less than 20,000 reads/counts detected or more than 0.05% of mitochondrial genes detected) were excluded. The cell types were annotated through an unbiased deep-learning model based on previously published retinal single-cell expression data (Clark et al., 2019) using scDeepSort version 1.0. Differential gene expression analyses were performed using Scanpy (Wilcoxon signed-rank test) or MAST version 1.24.042.

### Acknowledgements
We thank Michel Cayouette, Fei Chang and David Picketts for comments on a previous version of this manuscript. We thank Alena Kalinina for supporting scRNA-seq experiments. We thank Katia Georgopoulos and Toshimi Yoshida for sharing the *Chd4^Flox* mice. We thank Zev Gartner, Chris McGinnis, David Cook and Barbara Vanderhyden for generously providing Multi-seq protocols, reagents and advice. We thank the Michael Dyer and Seth Blackshaw labs for providing datasets upon which this study relied. For scRNA-seq and CUT&RUN-seq experiments, we also thank Katayoun Sheikheleslamy, Caroline Vergette and Pearl Campbell from the Stemcore Molecular Biology Core facility, as well as Chris Porter from the OHRI Bioinformatics Core Facility. We thank Chloë van Oostende-Triplet and the Cell Biology and Image Acquisition Core Facility, and the staff of the uOttawa Animal Care and Veterinary Service. We thank members of the Mattar and Picketts labs for their ongoing support and input.

### Competing interests
The authors declare no competing or financial interests.

### Author contributions
Conceptualization: S.S., P.M.; Formal analysis: J.A.L.F.; Funding acquisition: P.M.; Investigation: S.S., S.M., J.A.L.F., P.M.; Methodology: S.S., J.A.L.F.; Project administration: P.M.; Supervision: P.M.; Visualization: J.A.L.F.; Writing – original draft: S.S., P.M.; Writing – review & editing: S.S., S.M., J.A.L.F., P.M.

### Funding
This work was generously supported by the Canadian Institutes of Health Research (CIHR) Operating Grants (PJT-166032 and PJT-166074), as well as the New Frontiers in Research Fund (NFRFT-2022-00327). S.S. was generously supported by a David M. Shillito Scholarship in Ophthalmology Research. The project was also supported by an infrastructure grant from the Canada Foundation for Innovation for confocal microscopy (JELF 37688). P.M. gratefully holds the Gladys and

Lorna J. Wood Chair for Research in Vision. Open Access funding provided by the University of Ottawa. Deposited in PMC for immediate release.

**Data and resource availability**
ATAC-seq and CUT&RUN-seq datasets have been deposited in GEO database under accession number GSE266039. scRNA-seq data have been deposited in GEO database under accession number GSE300175. All other relevant data and details of resources can be found within the article and its supplementary information.

**Peer review history**
The peer review history is available online at https://journals.biologists.com/dev/lookup/doi/10.1242/dev.204697.reviewer-comments.pdf

**Special Issue**
This article is part of the Special Issue 'Lifelong Development: the Maintenance, Regeneration and Plasticity of Tissues', edited by Meritxell Huch and Mansi Srivastava. See related articles at https://journals.biologists.com/dev/issue/152/20.

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
