## [Peer Review File · Development (Cambridge, England)]

Chd4 remodels chromatin to control retinal cell type specification and lineage termination

Sujay Shah, Suma Mediseti, José Alex Lourenço Fernandes and Pierre Mattar

DOI: 10.1242/dev.204697

Editor: Francois Guillemot

Review timeline

Original submission: 3 February 2025

Editorial decision: 18 March 2025

First revision received: 28 July 2025

Accepted: 21 August 2025

Original submission

First decision letter

MS ID#: dev.204697

MS Title: Chd4 remodels chromatin to control retinal cell type specification and lineage termination

Authors: Sujay Shah; Suma Mediseti; José Alex Lourenço Fernandes; Pierre Mattar

Article Type: Research Article

Dear Dr Mattar,

I have now received the reports of two referees on your manuscript, and I have reached a decision. The referees' comments are appended below, or you can access them online: please go to:

As you will see, the two referees express great interest in your work but they also have some significant and convergent criticisms and recommend a substantial revision of your manuscript before we can consider publication. In particular, they both recommend that you strengthen the analysis of the retinal Chd4 mutant phenotypes by changing how some of the data is quantified, and also that you revise the discussion section. If you are able to revise the manuscript along the lines suggested, which may involve further experiments, I will be happy receive a revised version of the manuscript. Your revised paper will be re-reviewed by one or more of the original referees, and acceptance of your manuscript will depend on your addressing satisfactorily the reviewers' major concerns. Please also note that Development will normally permit only one round of major revision. If it would be helpful, you are welcome to contact us to discuss your revision in greater detail. Please send us a point-by-point response indicating your plans for addressing the referees' comments, and we will look over this and provide further guidance.

Please attend to all of the reviewers' comments and ensure that you clearly highlight all changes made in the revised manuscript. Please avoid using 'Tracked changes' in Word files as these are lost in PDF conversion. I should be grateful if you would also provide a point-by-point response detailing how you have dealt with the points raised by the reviewers in the 'Response to Reviewers' box. If you do not agree with any of their criticisms or suggestions please explain clearly why this is so.

Reviewer 1

SUMMARY OF THE ADVANCE MADE IN THIS PAPER AND ITS POTENTIAL SIGNIFICANCE TO THE FIELD

This is an interesting and impactful manuscript from Shah and colleagues about the effects of deleting the broadly expressed epigenetic modifier Chd4 from the developing mouse retina. The authors use a conditional knock out approach to remove Chd4 in retinal progenitor cells and then primarily examine postnatal phenotypes using histology and 'omics approaches (CUT&RUN, scRNA-seq, and ATAC-Seq). They find that loss of Chd4 alters cell proportions in the retina in a complex fashion and changes chromatin accessibility. Interestingly, accessibility changes did not correlate well with gene expression differences, but rather was more closely associated with Chd4 bound regions. They found that some Chd4 mutant cells did not properly exit the cell cycle at the end of normal development- a striking phenotype. Along these lines, they generated data suggesting that Chd4 mutant cells have a modestly shifted temporal identity profile. Overall, their data will be an important resource and contribution to the field.

SUGGESTIONS TO AUTHORS

Enthusiasm for this manuscript is high. Key strengths include logical experiments that generated high quality data, which will be both interesting and useful to the field. On the other hand, challenges attributing a mechanism to the observed cell fate demographic changes make interpretation difficult, reducing impact. As detailed below, I believe that these problems can be addressed relatively easily.

MAJOR CONCERNS:

1. It is often difficult to appreciate fate changes and their causation. For example, instances where the total number of cells is different between control and Chd4 cKO conditions (e.g., P15 timepoints), the authors' use of percentage calculations to assess fate changes is problematic. This is a significant issue in figure 3. The P15 retinas are far thinner, so an increase in a population percentage could be because the numerator (cell fate marker) went up or the denominator (total cells) went down. This type of problem would also tend to dampen the magnitude of populations that decrease (e.g., cones). As such, the authors should quantify their P15 timepoints based on retinal width for normalization, as they do in figure 1, etc. This will reveal whether the number of a given cell type formed during development changed or if it is an artifact of a changing denominator. The former would support a cell fate alteration mechanism. It is possible that these new results will change some of the conclusions made in this manuscript. If so, revisions to the discussion will also be needed.
2. The fate shift towards amacrine cells is not deeply supported. The increase in PAX6+ cells likely reflects amacrine cells, but also could be labeling horizontals (unknown change) and glia/progenitors (increased). The Pax6 quantification also suffers from the shrinking denominator problem mentioned above. This amacrine cell argument would be bolstered by a quantification of the AP2A staining in supplemental figure 5 and of the Sox2+ cholinergic amacrine cells in figures 5 and 7 (again normalized by width, rather than percentage). Alternatively, stains for amacrine cells could be done at several timepoints throughout development, especially embryonic stages, to detect fate changes versus other mechanisms, like cell survival (see below).
3. The fate shift away from rods is hard to appreciate. While the EdU birthdating suggests that rods are formed less often in the postnatal retina (figure 5), it is also possible that these rods do not survive well (matching the loss of ONL thickness). This alternative explanation is supported by the lack of an apparent ONL thickness change at P5, a time when nearly all of the rods have formed (particularly centrally). This cell death effect should be considered in the discussion of the data. Alternatively, a more systematic examination of rod photoreceptor formation prior to P8 could be completed to better discriminate between fate choice and cell survival effects.
4. In general, the idea of cell death is not explored deeply as a major mechanism for the cell fate phenotypes seen in Chd4 cKO retinas. The effect on cone survival is conspicuous and discussed by the authors (but see below) while the potential effects on rod survival are pointed out above. Moreover, it is possible that the increase in RGCs is due to a reduction in normal apoptosis (pruning) during development in Chd4 mutants. To strengthen the manuscript, the authors should deeply consider and discuss how differential cell survival could explain their findings.

5. The authors claim that they "...observed a surprisingly straightforward "temporal-looking" shift in neuron production...". This statement is not well supported by their data. There are more RGCs and likely amacrine cells, yet other early born cell types are either not studied (horizontal cells) or unchanged (cones) at the end of their normal genesis period. Similarly, it is unclear whether rods are formed less or survive poorly. Late born bipolar cell numbers are modestly increased. Thus, the mutant phenotype manifests as an increase on the extreme temporal edges, not a progressive shift (early or late). The gene expression profiling data in figure 5 does suggest that there is a slight shift of mutant cells towards a later temporal identity, but this shift can explain only a subset of the phenotypes. Overall, there are multiple ways that Chd4 could be acting in the retina to regulate cell identity, competence, cell cycle exit, temporal identity, and survival. This is a fascinating interplay of phenotypes and makes for a compelling and complex discussion. The authors should consider emphasizing this complexity of phenotypes rather than oversimplifying to fit a shifting temporal identity model. To better support their temporal identity shift model, the authors would need to show how Chd4 mutant cell fate changes at many timepoints, especially embryonically. This represents a lot of work and is likely outside the scope of this manuscript.

MINOR CONCERNS:

1. There are few markers used to identify photoreceptors, bipolar cells, and glia. Should Chd4 loss affect marker expression, misidentification of cell types could occur. For example, the loss of cone arrestin may reflect a lack of maturation rather than death of these cells. There are also no markers used to distinguish between progenitors and glia. Hence, there may be extra progenitors that fail to differentiate rather than an increase in the formation of glia. While not strictly necessary, the use of additional markers would significantly strengthen the manuscript. Along these lines, a deeper analysis with multiple markers would help clarify the identity of mislocalized cell nuclei in the outer nuclear layer.

2. Why do only a subset of cells fail to exit the cell cycle or become glia? Some discussion on why this partial phenotype occurs would help readers better understand the manuscript.

3. The viral experiments to delete Chd4 starting at birth are poorly described in the results section. As this makes it hard to understand, please explain the experimental parameters in the results section in more detail. The cell fate conclusions about this experiment are not supported by supplemental figure 12. In particular, none of the fate changes are denoted as being significantly different than the control condition. Is this an omission of the stats or are there really no significant differences? If the former, please update the figure. If the latter, the conclusions mentioned in the results section are not supported and should be updated to reflect the lack of a phenotype from the viral-mediated Chd4 knockout.

4. Despite forming extra RGCs and amacrine cells, the inner plexiform layer is very thin. Is this a deficit in dendrite formation caused by mutating Chd4? Some other mechanism? A discussion of why this layer is so thin would improve the manuscript.

5. In several places in the figures, it is unclear whether the authors are staining for Cre recombinase or YFP to detect their transgene. In figure 4I-K, it looks like some staining is cytoplasmic YFP while other panels look like nuclear-localized Cre recombinase. Please label the panels throughout the paper to reflect the antibody used to detect the transgene. In addition, there is no information in the methods about what Cre antibody was used. Please update the methods accordingly.

6. It is unclear why the model figure lacks horizontal cells and what the different colored arrows denote. Please update accordingly and consider the model design in light of the comments above.

7. Please consider revising the discussion point about Nfi genes to include the bipolar cell phenotypes that were observed in triple mutants.

8. Please also consider citing Pequegnot et al, 2003 (PMID: 14517994) about developmental cell death. Note that amacrine cell death peaks earlier than that of bipolar cells. Please consider citing Carter-Dawson and LaVail, 1979 (PMID: 500859) concerning photoreceptor birthdating in mice.

Reviewer 2

SUMMARY OF THE ADVANCE MADE IN THIS PAPER AND ITS POTENTIAL SIGNIFICANCE TO THE FIELD

Overall, this study reports a number of interesting retinal phenotypes related to Chd4 function using both cell biological analysis and -omics approaches. The experiments are well done and present novel findings within the retina. Some of the results were predictable given previous analysis of Chd4 function in other parts of the nervous system, but still extend significantly our understanding of this gene's function. As the Chd4 protein may function in a number of different pathways and those relevant pathways for the observed phenotypes are not identified in this study, the mechanistic conclusions however are limited. But, as a first report of the mutant phenotype in the retina, it identifies some novel phenotypes and associated datasets that will be of significant interest to the field.

SUGGESTIONS TO AUTHORS

Major Suggestions

One of the major issues is the assessment of cell fate changes. It is obvious that there is a phenotype with this mutant, but it is not entirely clear from the data that is presented whether this is due to cell fate changes or ectopic marker expression and there are some confounding effects of layering defects and presumed cell death that are taking place as well. Some of the issues in this area that should be addressed are as follows:

1) Do percentages take into account moving changes that affect the denominator? It is stated that there is no change in Hoechst counts, but there seems like a 15% drop for the P2 and P8 timepoints. The major issue is at P15 timepoint where the authors describe a significant reduction in cell counts. The increase in Pax6 percentage could just be the mathematical result of a smaller total cell denominator due to the drop in rods and other cell types that the authors describe at this time. The quantification of bipolar cells at P8 might suffer the same issue as the ONL is thinner and the cell counts for the total retina are fewer. Perhaps these cell numbers can be calculated by just determining the absolute number of these cells in a given area of the retina without calculating a percentage based on the total number of cells in the retina?

2) At the early timepoints it looks like there is a change in amacrine cell numbers (for example the number of GFP-negative cells immediately below the RPC layer at P0 and P2 in Fig 2D-F where amacrine cells would be found, is noticeably thin in the floxed allele retina - is it possible these are somehow becoming RGCs? Or are the amacrine cells not maintained in their layer and becoming more interspersed with the RGC layer, which makes the RGC layer look bigger? This loss of the amacrine cells below the Chx10+ RPCs can also be observed in FigS3. Staining with Tfp2a antibodies that are specific for amacrine cells and not RGCs could resolve this - this was done in Fig S5, but not quantified - this should be done. In Figure 3 the authors use Pax6, but the INL is not well formed and there seems like there could be issues of RGC INL segregation. And Pax6 does not seem to be labeling a lot of the cells in the INL of the controls (Fig 3C) that one would think are probably amacrine cells so the Pax6 staining is not going to allow for an accurate count of amacrine cells.

The single cell analysis, Chd4 CUT&RUN and ATACSeq datasets are well done and also presented nicely. They show some very interesting effects such as increased open chromatin at regions not bound by Chd4, leaving it an open question as to how those changes arise. That is probably beyond the scope of this study, though if there was some indication of what these peaks represented (specific elements related to RPC-specific gene expression, repressor sequences, etc) that would be value added. The single cell analysis shows that there seems to be a striking effect on the UMAP plot of the mutant in a nicely controlled experiment, but it is a little hard to understand what the take home is as all of the cells seem to be shifted but there are no clear differences in cell population numbers noted. The authors targeted an early postnatal timepoint to better understand Chd4 gene function at the time it is acting, but the difficulty in assessing cell types because of ambiguities in identity make it difficult to interpret. It is not clear if there are detectable differences in the numbers of specific cell types such as rods or RPCs. Can cell percentages for these cells be included so as to observe whether there are changes in rod formation?

One of the retina biology take-homes of the study is that Chd4 regulates RPC to rod transitions in the postnatal period. The study does provide some solid evidence that there are indeed effects on RPCs (like developmental persistence past their normal time window) and also on rod formation (less rods formed). But because the NuRD complex could be working with many different transcription factors and involved in separate transcriptional events, it may be that the effects are not tied mechanistically together. Perhaps Chd4 is involved in repressing the RPC state through effects on specific genes, but maybe it separately also keeps some genes turned off in rods that are toxic to them. The authors present a reasonable model and should keep it as a model - but without a deeper mechanistic understanding of how Chd4 is exerting these effects it would be best to discuss other possibilities for the phenotypes that are observed.

Minor Suggestions

It would help to have line numbers for reviewing.

The original Chx10 transgenic as described in Rowan and Cepko 2004 refers to the encoded fluorescent reporter as EGFP, but this manuscript refers consistently to it as EYFP. If the authors have a justification for this reference they should make the case, but if not, EGFP should be used.

In Fig S1 the single cell plots don't make it obvious to a reader the expression differences of the Chd3-5 genes. For example, neurogenic cells don't look that different in terms of shading compared to the other cell populations, but the boxplot looks very different. Inclusion of some other gene plots may help to make a comparison here or perhaps the scaling can be adjusted to make the differences and the general expression more obvious?

In Figure 2A some indication of the indicated Chd4 band size based on a ladder would be important to show that this is likely the predicted Chd4 band (and an indication of what that predicted band size is).

In figure 2, is the Chx10 channel [which should be labeled EGFP or GFP^{Cre} (to represent the fusion protein) to be more direct in what is being observed] being shown in the Chd4 Hoechst picture? The retina presumably lacks the GFP signal because this animal doesn't have the transgene, but would be nice to show it here, or if the green channel is being shown to note that it is being visualized.

Consistency in labeling and description of the actual target that is being observed should be done when visualizing the GFPChx10iresCre transgene (GFP fluorescence, GFP antibody, Cre, Chx10 antibody). In Figure 2 "Chx10 is used", in figure 3 it is "Cre" to "Chx10 Cre" in fig s4- if they are in fact different this should be noted. A consistent method for labeling should be applied for all the figures that assess a marker expressed by the transgene.

The authors indicate that they did not observe any mosaicism as is commonly seen for the Chx10 transgene - it could be useful to show a whole retina view to show this. Later on when discussing the single cell data there is a reference to a possible mosaicism in one of the samples, so an image could bolster the case that there isn't. Though further information should be provided for the sample in the single cell experiment as noted in another part of this review.

The Sox2 staining in Fig 7I and J looks like a lot of it is something else and not actually Sox2 which should be nuclear and is the band of cells at the top of the INL. The very strong Sox2 signal in cells outside of the MG layer and with cytoplasmic staining is not consistent with these being Mueller glia and yet seems to be increasing in the mutant. This may be throwing off some of the counts of actual Mueller cells - were they counted? Can another more specific marker be used? Perhaps sox9, p27, or Lhx2?

The viral tracing does not seem to have the sample size to be able to conclude statistical significance - the authors don't do so, but it is best to either increase the sample size to reach this threshold or not include the data.

For the single cell analysis there is a mention that "...these changes are observed despite probable mosaicism in one of the Chd4 cKO samples...". It is not clear what is being referred to here. This point should be expanded as it would be important to know what the indication of this potential

confounding effect is and what is its origin. Related is that it would be beneficial to show the individual UMAPs of each biological sample as its somewhat difficult to see each one - this could be done in supplemental material. It would also be useful to note the sex of each mouse, which could presumably be determined from gene expression such as *xist*.

First revision

Author response to reviewers' comments

We thank the Reviewers for all the time that they devoted to reviewing our manuscript, and for pointing out ways to improve it. We have done our best to address the identified concerns, and we hope that the Reviewers will agree that our revisions have adequately addressed the issues that they identified. We think that the suggestions have led to marked improvements in the quality of the manuscript. We have also introduced changes to the manuscript in order to meet space requirements, including moving some elements of the text and figures into the Supplemental Section.

Below, our responses to Reviewer comments are in blue, and changes quoted from the new version of the manuscript are indicated in red.

Reviewer 1: SUMMARY OF THE ADVANCE MADE IN THIS PAPER AND ITS POTENTIAL SIGNIFICANCE TO THE FIELD

This is an interesting and impactful manuscript from Shah and colleagues about the effects of deleting the broadly expressed epigenetic modifier *Chd4* from the developing mouse retina. The authors use a conditional knock out approach to remove *Chd4* in retinal progenitor cells and then primarily examine postnatal phenotypes using histology and 'omics approaches (CUT&RUN, scRNA-seq, and ATAC-Seq). They find that loss of *Chd4* alters cell proportions in the retina in a complex fashion and changes chromatin accessibility. Interestingly, accessibility changes did not correlate well with gene expression differences, but rather was more closely associated with *Chd4* bound regions. They found that some *Chd4* mutant cells did not properly exit the cell cycle at the end of normal development- a striking phenotype. Along these lines, they generated data suggesting that *Chd4* mutant cells have a modestly shifted temporal identity profile. Overall, their data will be an important resource and contribution to the field.

We thank the Reviewer for these very generous comments.

SUGGESTIONS TO AUTHORS

Enthusiasm for this manuscript is high. Key strengths include logical experiments that generated high quality data, which will be both interesting and useful to the field. On the other hand, challenges attributing a mechanism to the observed cell fate demographic changes make interpretation difficult, reducing impact. As detailed below, I believe that these problems can be addressed relatively easily.

MAJOR CONCERNS:

1. It is often difficult to appreciate fate changes and their causation. For example, instances where the total number of cells is different between control and *Chd4* cKO conditions (e.g., P15 timepoints), the authors' use of percentage calculations to assess fate changes is problematic. This is a significant issue in figure 3. The P15 retinas are far thinner, so an increase in a population percentage could be because the numerator (cell fate marker) went up or the denominator (total cells) went down. This type of problem would also tend to dampen the magnitude of populations that decrease (e.g., cones). As such, the authors should quantify their P15 timepoints based on retinal width for normalization, as they do in figure 1, etc. This will reveal whether the number of a given cell type formed during development changed or if it is an artifact of a changing denominator. The former would support a cell fate alteration mechanism. It is possible that these

new results will change some of the conclusions made in this manuscript. If so, revisions to the discussion will also be needed.

This issue was flagged by both Reviewers, and we agree completely. We have therefore added the absolute cell counts to the paper (new Fig. S5). At P15, we do not see significant changes in absolute cell type numbers in *Chd4* cKO versus wild-type or cHets at P15, except that rods and cones are significantly reduced. We now mention these results in the text. E.g. pp. 7:

These changes were observed in proportional counts, although not in absolute numbers (Fig. S5).

As the Reviewer states above, we think that the key issue is whether cell type production is altered *during development* or whether changes are instead driven post-hoc by cell death. Indeed, cell death is known to re-establish the proper number of several retinal cell types if they are overproduced (Brzezinski et al., 2010; Dyer and Cepko, 2000), potentially masking changes in cell type production. For this reason, we previously examined the *Chd4* cKO at P8. However, while we had counted Sox2+ cells and Otx2+ bipolars, we had not formally measured most cell types, including photoreceptors.

To address cell type production prior to P15, we have added new analyses and cell counts from P5, P6, and P8 stages. At P5, we confirm that overall cell numbers are not significantly changed between the cKO versus controls (Fig. 5D), although there is still a trend towards reduced numbers pointed out by Reviewer 2 (similar to P8 counts in Fig. 2K). We therefore additionally present absolute counts for key P5 and P8 data in Fig. S5. At P5, we already see a significant reduction in the numbers of Otx2+/Nr2e3+ rod photoreceptors - both in absolute and relative terms (Fig. 5E, Fig. S5J). At P8, we have added counts of Ki67+ RPCs, confirming a highly significant accumulation of RPCs in the *Chd4* cKO (Fig. 5R). Thus, differentiating rods are significantly reduced and RPCs are increased at these stages. The observed increases (absolute and proportional) in proliferating RPCs are very difficult to explain via a cell death -driven mechanism.

However, although Hoechst counts in Fig. 2 suggest that significant dieback in the *Chd4* cKO mainly occurs between P8 and P15, and although we had previously examined cell death at P0 via TUNEL stainings, the precise onset of the cell death that reduces the cell count by P15 was unclear. In new experiments, we performed stainings for activated Caspase 3 at P5, and observed a significant elevation of apoptosis. This elevation in cell death is not (yet) sufficient to alter the overall cell numbers (new data shown in Fig. 5D). However, we ultimately agree with the Reviewers' point that apoptosis could be driving changes in absolute cell numbers for particular fates - even prior to cell death significantly altering the overall cell count. We have therefore expanded the discussion to further develop this point (pp. 14):

While distortions in cell type composition were observed prior to the significant cell loss that occurs between P8 and P15, alterations cell death may nonetheless be responsible for the observed shifts in cell type composition. Increases in early-born fates - including RGCs and cholinergic amacrine, occurred without a complementary decrease in alternative fates. While early hyperproliferation might explain the expansion of early-born fates, RGC apoptosis might instead be reduced. Many supernumerary RGCs die at perinatal stages, but there is an additional wave of embryonic RGC cell death that we did not investigate (Farah, 2006; Pequignot et al., 2003). At birth, rod and cone numbers are not altered in the *Chd4* cKO, but they become significantly reduced by P5. These reductions corresponded with a significant elevation in apoptosis. While it is therefore difficult to exclude the notion that photoreceptors are selectively eliminated at late developmental stages, the concomitant expansion of cycling RPCs in the *Chd4* cKO suggests that in addition to apoptotic loss, RPC self-renewal is increased, which we think would likely undermine rod production at late stages.

2. The fate shift towards amacrine cells is not deeply supported. The increase in PAX6+ cells likely reflects amacrines, but also could be labeling horizontals (unknown change) and glia/progenitors (increased). The Pax6 quantification also suffers from the shrinking denominator problem mentioned above. This amacrine cell argument would be bolstered by a quantification of the AP2A staining in supplemental figure 5 and of the Sox2+ cholinergic amacrine cells in figures 5 and 7 (again normalized by width, rather than percentage). Alternatively, stains for amacrine cells could be done at several timepoints throughout development, especially embryonic stages, to detect fate

changes versus other mechanisms, like cell survival (see below).

In the current version of the manuscript, we added counts for *Tfap2a* at P8 (Fig. 5K), which did not reveal a significant increase in amacrine cells. However, we additionally re-examined our *Sox2* stainings, and have added counts of cholinergic (starburst) amacrine cells, which were significantly increased (Fig. 5O - basal cells). However, in terms of the magnitude of the effect size, we agree with the Reviewer that the observed changes are modest. We have therefore adjusted the text and summary figure, so that it only refers to differences in amacrine numbers with respect to “early-born” amacrine cells, and not the whole population. We also clarified that the P15 *Pax6* counts presented in Fig. 2 are only a measure of amacrine cells (pp. 7).

As a partial measure of amacrine cells, we counted *Pax6*⁺ cells within the INL. *cKO* retinas displayed a slight increase in INL amacrine cells when compared to wt (Figure 3C, D). These changes were observed in proportional counts, although not in absolute numbers (Fig. S5).

Finally, to address horizontal cells, we performed co-stainings for *Lhx1* and calbindin, and we counted *Lhx1*⁺ cells in 300 micron-wide bins at P5. Counts indicate that the overall density of horizontal cells is not altered in the *Chd4* *cKO*. These data are presented in Fig. S6. We also added horizontal cells to our summary figure as requested.

3. The fate shift away from rods is hard to appreciate. While the EdU birthdating suggests that rods are formed less often in the postnatal retina (figure 5), it is also possible that these rods do not survive well (matching the loss of ONL thickness). This alternative explanation is supported by the lack of an apparent ONL thickness change at P5, a time when nearly all of the rods have formed (particularly centrally). This cell death effect should be considered in the discussion of the data. Alternatively, a more systematic examination of rod photoreceptor formation prior to P8 could be completed to better discriminate between fate choice and cell survival effects.

As mentioned above, to further address the point, we now add *Nr2e3* rod cell counts at P5 (Fig. 5E, absolute counts: Fig. S5J). These are significantly reduced. As mentioned above, since we also detected significantly increased apoptosis at this stage, we now discuss the possibility that cell death selectively targets photoreceptors to reduce rod and cone counts postnatally in the discussion (pp. 14) as described above in response to comment #1.

We think that the concomitant expansion in cycling RPCs argues (but does not prove) that the production of rods is likely reduced at late stages. Late-stage EdU birthdating also argues that rods are underproduced, since we did not see EdU⁺ rods in P8 to P15 birthdating experiments in the *cKO* (Fig. S10), although EdU⁺ rods are generated in controls - albeit only at the peripheral margins of the retina. However, it remains possible that late-born rods are selectively killed via apoptosis. We therefore do not explicitly make this latter argument in the text.

4. In general, the idea of cell death is not explored deeply as a major mechanism for the cell fate phenotypes seen in *Chd4* *cKO* retinas. The effect on cone survival is conspicuous and discussed by the authors (but see below) while the potential effects on rod survival are pointed out above. Moreover, it is possible that the increase in RGCs is due to a reduction in normal apoptosis (pruning) during development in *Chd4* mutants. To strengthen the manuscript, the authors should deeply consider and discuss how differential cell survival could explain their findings.

We thank the Reviewers for prompting us to look at apoptosis more carefully, and apologize for not properly addressing cell death in the prior version. With respect to lack of apoptosis driving the RGC expansion, we previously counted TUNEL positive cells at P0, but do not see a reduction in apoptosis in the *Chd4* *cKO*. However, while P0 is around the peak of developmental RGC death, as the Reviewers both pointed out, there is an earlier wave of RGC apoptosis that occurs at embryonic stages that we have not examined. We now address the potential scenario that lack of apoptosis might drive the expansions in early-born cell types observed in the *cKO*.

In addition to performing further analyses of cell death in the *cKO*, we have added a paragraph to the discussion as described above in our response to point #1.

5. The authors claim that they “...observed a surprisingly straightforward “temporal-looking” shift in

neuron production...". This statement is not well supported by their data. There are more RGCs and likely amacrine cells, yet other early born cell types are either not studied (horizontal cells) or unchanged (cones) at the end of their normal genesis period. Similarly, it is unclear whether rods are formed less or survive poorly. Late born bipolar cell numbers are modestly increased. Thus, the mutant phenotype manifests as an increase on the extreme temporal edges, not a progressive shift (early or late). The gene expression profiling data in figure 5 does suggest that there is a slight shift of mutant cells towards a later temporal identity, but this shift can explain only a subset of the phenotypes. Overall, there are multiple ways that *Chd4* could be acting in the retina to regulate cell identity, competence, cell cycle exit, temporal identity, and survival. This is a fascinating interplay of phenotypes and makes for a compelling and complex discussion. The authors should consider emphasizing this complexity of phenotypes rather than oversimplifying to fit a shifting temporal identity model. To better support their temporal identity shift model, the authors would need to show how *Chd4* mutant cell fate changes at many timepoints, especially embryonically. This represents a lot of work and is likely outside the scope of this manuscript.

We agree completely with the Reviewer. Our original paragraph went on to ‘shoot down’ the simple “temporal- looking” interpretation, since it does not fit our data. So, the original sentence flagged above was originally meant as a straw man. However, we agree that this approach can lead the reader to misinterpret our conclusions. We have therefore restructured this paragraph to follow more closely the Reviewer’s interpretations above (pp. 14):

Although temporal transitions play a critical role in diversifying neural progenitor lineages, the underlying molecular mechanisms are poorly understood. Here, we addressed the role of nucleosome remodelling in regulating the chronology of cell type production from retinal progenitors. We hypothesized that *Chd4* -dependent chromatin remodelling would be required for the competence transition between embryonic and postnatal modes of neurogenesis. *Chd4* cKOs accordingly exhibited increases in early-born RGCs and amacrines, and later-born rods were drastically decreased (Fig. 8A, B), initially suggesting a prolongation of early competence. However, this proved not to be the case. Indeed, scRNA-seq profiling suggested that *Chd4* cKOs were instead slightly accelerated in their temporal state. Moreover, we observed that cycling RPCs persisted beyond their normal developmental window and differentiated into glia much later than controls. Taken together, these data suggest that *Chd4*-dependent nucleosome remodelling regulates the temporal transition that terminates the retinal lineage, but does not control earlier competence transitions.

MINOR CONCERNS:

1. There are few markers used to identify photoreceptors, bipolar cells, and glia. Should *Chd4* loss affect marker expression, misidentification of cell types could occur. For example, the loss of cone arrestin may reflect a lack of maturation rather than death of these cells. There are also no markers used to distinguish between progenitors and glia. Hence, there may be extra progenitors that fail to differentiate rather than an increase in the formation of glia. While not strictly necessary, the use of additional markers would significantly strengthen the manuscript. Along these lines, a deeper analysis with multiple markers would help clarify the identity of mislocalized cell nuclei in the outer nuclear layer.

The comment that Müllers fail to differentiate in the cKO proved to be completely correct. We are therefore very grateful to the Reviewer for submitting this criticism, and feel strongly that addressing this point has greatly helped to improve the paper. We previously performed scRNA-seq in part to help resolve these sorts of issues, but the chosen timepoint is too early to address the Reviewer’s comment above. We have therefore added several new markers (as well as formal measurements) to the current version of the manuscript (as highlighted in green below) - focusing our attention on later postnatal stages.

In particular, we added *Rlb1* stainings as a specific marker of Müller glia (Fig. 5S, T, Fig. S11). *Ki67* specifically marks RPCs but not Müllers. Along with new *Ki67* counts, *Rlb1* stainings confirm that at P6-P8, RPCs persist and Müller cells do not differentiate on schedule in the *Chd4* cKO. We hope that the Reviewer will agree that the inclusion of these markers helps to address the concerns raised above.

RPCs: GFP+ (i.e. Vsx2+ cells; P0 count Fig. 4E); Ki67 (P2 count Fig. 4I-L; new P8 count Fig. 5R); EdU incorporation (P0-P2 count; Fig 4I-L; new P8-P15 experiment Fig. S10)

RGCs: Brn3a (P0 counts: Fig. 4A-D); Brn3b (Fig. S6); Rbpms (P15 counts: Fig. 3A, B; Fig. S6); Rxrg (in GCL Fig. 4F, G), new neurofilament (in nerve fiber layer, Fig. S6).

Amacrine: Pax6 (counts from INL only: Fig. 3C, D); Sox2 (new P8 counts from basal INL and GCL: Fig. 5R); Tap2a (new P8 count Fig. 5R; Fig. S6)

Horizontals: New Lhx1/calbindin co-stainings at P6, Lhx1 count at P5; new neurofilament (in outer plexiform layer at P8, Fig. S6).

Cones: Cone arrestin counts at P15 (Fig. 3E, F); Rxrg (in ONBL: P0 count in Fig. 4F-H); new peanut agglutinin wholemount staining at P21 (Fig. S6)

Rods: Cone arrestin-negative ONL cells at P15 (Fig. 3E, F); Otx2+Nr2e3+ cells at P5 (Fig. 5A-B)

Bipolars: GFP+/Otx2+ cells (P15 counts Fig. 3G, H; P8 counts Fig. 7G-K)

Müllers: Rbp1 (new P6 staining Fig. 5R, S, new P6, P8, P15 stainings Fig. S11)

2. Why do only a subset of cells fail to exit the cell cycle or become glia? Some discussion on why this partial phenotype occurs would help readers better understand the manuscript.

Due to the issues with discriminating between RPCs and Müllers discussed above, this is a challenging question to address. Adding counts that would track this with precision proved to be too challenging for us to be able to achieve, but we did perform a timecourse analysis with Rbp1 (new Fig. S11). By P15, Rbp1 staining is widespread in the *Chd4* cKO (Fig. S11). In data that we did not include in the paper, we also examined Ki67 at later stages. At P15, we still see a few Ki67+ cells, but only ~10 cells per section are labelled. We now state (pp. 10):

In the cKO, Rbp1 was little expressed at P6 and P8, but was expressed by P15 (Fig. S11). Taken together, these data demonstrate that RPCs fail to differentiate into quiescent glia on schedule and accumulate in the *Chd4* cKO, but eventually differentiate.

3. The viral experiments to delete *Chd4* starting at birth are poorly described in the results section. As this makes it hard to understand, please explain the experimental parameters in the results section in more detail. The cell fate conclusions about this experiment are not supported by supplemental figure 12. In particular, none of the fate changes are denoted as being significantly different than the control condition. Is this an omission of the stats or are there really no significant differences? If the former, please update the figure. If the latter, the conclusions mentioned in the results section are not supported and should be updated to reflect the lack of a phenotype from the viral-mediated *Chd4* knockout.

For these experiments, we could not perform statistical analyses due to the loss of one of the side-by-side control experiments (i.e. n=2 for the experiment). As a result, we opted to simply remove these experiments. We apologize to both Reviewers for this issue.

The prior experiments suggested that the clone size distribution generated by cKO RPCs was unaffected, but based on our other results, we now think that this result might reflect a balance between prolonged proliferation and increased cell death at late stages. Since the Cre in these experiments was introduced at P0, we think that more investigation would be needed to clarify whether the proportional changes seen in rod and Müller numbers arises via the same mechanism as for the Chx10-Cre-GFP line.

4. Despite forming extra RGCs and amacrine cells, the inner plexiform layer is very thin. Is this a deficit in dendrite formation caused by mutating *Chd4*? Some other mechanism? A discussion of why this layer is so thin would improve the manuscript.

The Reviewer is absolutely right that the lamination of the retina is overtly disrupted/delayed in the *Chd4* cKO - both for the inner and outer plexiform layers. In new calbindin and neurofilament stainings (newly added to Fig. S6), we accordingly see that horizontal cells exhibit defects in axodendritic processes in the outer plexiform layer. Neurofilament stainings in the nerve fiber layer revealed the presence of RGC axon bundles in the cKO, although these axons also looked disorganized, and few axons appeared to misproject away from the optic nerve head. Grossly, the optic nerve is present in the cKOs, although it is smaller in comparison to controls (not shown).

Moreover, work from Azad Bonni's lab in the cerebellum has shown that Chd4 is indeed required for the morphological differentiation and neurite outgrowth of cerebellar granule cells. We now discuss this briefly on pp.:15. We apologize for not elaborating further, but we are constrained in terms of the overall wordcount.

While *Ikzf1* and *Casz1* mutants have relatively mild phenotypes, Chd4 abrogation led to marked deficits in retinal lamination. Chd4 may be required for neurite outgrowth as shown previously in cerebellar granule neurons (Yamada et al., 2014). The greater severity of the Chd4 phenotype suggests that besides *Casz1*, additional transcription factors may interact with Chd4/NuRD in a stage-dependent fashion (Fig. 8C). Future work will be required in order to better understand the mechanisms that control retinal histogenesis.

To try to further address this question, we additionally re-examined DEGs and GO terms from the scRNA-seq dataset - both in RPCs, and in neurogenic cells and neurons. However, we did not observe an obvious signature associated with the outgrowth of axons or dendrites that might help explain the phenotype.

We also initially thought that the observed delay in Müller glia differentiation might conceivably contribute to lamination defects, since the separation between the ONL and INL occurs at around the time that Müllers differentiate. However, prior works have disrupted Müller differentiation without producing lamination defects (MacDonald et al., 2015; Poche et al., 2008). Moreover, defects in the separation of the GCL vs. ONBL/INL are apparent at embryonic stages - well before Müllers differentiate. As described above, we think that more investigation is needed in order to better understand this aspect of the cKO phenotype.

5. In several places in the figures, it is unclear whether the authors are staining for Cre recombinase or YFP to detect their transgene. In figure 4I-K, it looks like some staining is cytoplasmic YFP while other panels look like nuclear-localized Cre recombinase. Please label the panels throughout the paper to reflect the antibody used to detect the transgene. In addition, there is no information in the methods about what Cre antibody was used. Please update the methods accordingly.

We thank the Reviewer for catching this issue. We have now labelled all figures with "GFP" throughout the paper. We used epifluorescence to detect GFP in all experiments, except for Edu experiments, where we used an anti- GFP antibody.

6. It is unclear why the model figure lacks horizontal cells and what the different colored arrows denote. Please update accordingly and consider the model design in light of the comments above.

We have now added horizontal cells to the figure, and we now explain the meaning of the arrow colors in the legend. We additionally indicate the late-stage increase in apoptosis to the figure.

7. Please consider revising the discussion point about *Nfi* genes to include the bipolar cell phenotypes that were observed in triple mutants.

We now state (pp. 16):

Intriguingly, RPCs in *Nfia/b/x* triple mutants were previously shown to persist beyond the normal termination of neurogenesis, generating supernumerary rods, and underproducing bipolars and Müllers (Clark et al., 2019).

8. Please also consider citing Pequegnot et al, 2003 (PMID: 14517994) about developmental cell death. Note that amacrine cell death peaks earlier than that of bipolar cells. Please consider citing Carter-Dawson and LaVail, 1979 (PMID: 500859) concerning photoreceptor birthdating in mice.

We added the Pequegnot citation to the manuscript (along with Farah and Easter, 2005). In Farah and Easter, the peak of death of displaced amacrine (as determined via a birthdating strategy) was P6, but as far as we can see, other works have found little-to-no developmental death of amacrine in wild-types, although most studies (including Pequegnot) have not used cell-type markers. We have also added the Carter-Dawson and Lavail citation to the passages describing rod

photoreceptor birthdates.

Reviewer 2: SUMMARY OF THE ADVANCE MADE IN THIS PAPER AND ITS POTENTIAL SIGNIFICANCE TO THE FIELD

Overall, this study reports a number of interesting retinal phenotypes related to Chd4 function using both cell biological analysis and -omics approaches. The experiments are well done and present novel findings within the retina. Some of the results were predictable given previous analysis of Chd4 function in other parts of the nervous system, but still extend significantly our understanding of this gene's function. As the Chd4 protein may function in a number of different pathways and those relevant pathways for the observed phenotypes are not identified in this study, the mechanistic conclusions however are limited. But, as a first report of the mutant phenotype in the retina, it identifies some novel phenotypes and associated datasets that will be of significant interest to the field.

We thank the Reviewer for these positive comments.

SUGGESTIONS TO AUTHORS

Major Suggestions

One of the major issues is the assessment of cell fate changes. It is obvious that there is a phenotype with this mutant, but it is not entirely clear from the data that is presented whether this is due to cell fate changes or ectopic marker expression and there are some confounding effects of layering defects and presumed cell death that are taking place as well. Some of the issues in this area that should be addressed are as follows:

1) Do percentages take into account moving changes that affect the denominator? It is stated that there is no change in Hoechst counts, but there seems like a 15% drop for the P2 and P8 timepoints. The major issue is at P15 timepoint where the authors describe a significant reduction in cell counts. The increase in Pax6 percentage could just be the mathematical result of a smaller total cell denominator due to the drop in rods and other cell types that the authors describe at this time. The quantification of bipolar cells at P8 might suffer the same issue as the ONL is thinner and the cell counts for the total retina are fewer. Perhaps these cell numbers can be calculated by just determining the absolute number of these cells in a given area of the retina without calculating a percentage based on the total number of cells in the retina?

Again, we thank both Reviewers for these helpful comments. As mentioned above, we have now added absolute count data to the Supplemental section. We agree completely that the denominator issue flagged above completely distorts the cell-type numbers seen at P15, but we think that the changes seen in proportional numbers match the changes seen in counts at earlier stages (both in absolute and proportional terms).

2) At the early timepoints it looks like there is a change in amacrine cell numbers (for example the number of GFP-negative cells immediately below the RPC layer at P0 and P2 in Fig 2D-F where amacrine cells would be found, is noticeably thin in the floxed allele retina - is it possible these are somehow becoming RGCs? Or are the amacrine cells not maintained in their layer and becoming more interspersed with the RGC layer, which makes the RGC layer look bigger? This loss of the amacrine cells below the Chx10+ RPCs can also be observed in FigS3. Staining with Tfap2a antibodies that are specific for amacrine cells and not RGCs could resolve this - this was done in Fig 5, but not quantified - this should be done. In Figure 3 the authors use Pax6, but the INL is not well formed and there seems like there could be issues of RGC INL segregation. And Pax6 does not seem to be labeling a lot of the cells in the INL of the controls (Fig 3C) that one would think are probably amacrine cells so the Pax6 staining is not going to allow for an accurate count of amacrine cells.

We have now added counts for Tfap2a cells and Sox2+ cholinergic amacrine cells to Fig. 5, as described above.

We also initially wondered about whether amacrine cells and RGCs might exhibit mixed identities. To address this, we performed the co-stainings for Rbpms and Tfap2a mentioned by the Reviewer, but

did not observe co-expressing cells. These images are currently presented in Fig. S6 (previously Fig. S5). Moreover, scRNAseq analyses did not indicate a merging of RGC/amacrine fates. Although only a few RGCs were recovered, these cells clustered apart from all other cell types (red circle; note that all 22 of the detected RGCs came from cKO replicates, but that when we lowered our quality cutoffs, RGCs from wild-type cells clustered together with these RGCs). This was a very small cluster, so we did not provide images of marker expression in these cells, but typical RGC markers such as *Pou4f2* and *Rbpms* are highly expressed in these cells in accordance with expectations.

However, we agree completely that in the cKO, the lamination of the INL/GCL fails, leading to most amacrine cells becoming displaced. The very large size of the GCL at later stages reflects an increase in RGCs (including in absolute terms, Fig. S5H), as well as an increase in these displaced amacrine cells. However, we did not try to discriminate between INL and GCL amacrine cells, except at P8/P15, when we found that the inner plexiform layer was robust enough to make such distinctions reliable.

The single cell analysis, *Chd4* CUT&RUN and ATAC-seq datasets are well done and also presented nicely. They show some very interesting effects such as increased open chromatin at regions not bound by *Chd4*, leaving it an open question as to how those changes arise. That is probably beyond the scope of this study, though if there was some indication of what these peaks represented (specific elements related to RPC-specific gene expression, repressor sequences, etc) that would be value added.

To address this question, we performed motif footprinting using the TOBIAS algorithm on the ATAC-seq dataset to compare changes in transcription factor occupancy between control and cKO cells. These data are presented in Fig. 7H (previously Fig. 6H). The observation that peaks such as *Ctcf* are differential in the data suggests a major reorganization of the enhancer looping landscape of RPCs. In experiments not included in the manuscript, we have indeed confirmed that *Ctcf* recruitment to the genome is increased in the *Chd4* cKO using cut&run-seq (to be published elsewhere). We agree with the Reviewer that a detailed analysis genome remodelling needs a full study in order to determine how exactly *Chd4* might do this, but the existing data hint at a rearrangement of enhancer/promoter looping. Importantly, this sort of phenotype has already been reported in cerebellar granule neurons, which we mention in the paper (pp. 13).

Lastly, we performed footprinting analysis on accessible peaks in order to identify differential transcription factor occupancy using the TOBIAS algorithm (Fig. 7H) (Bentsen et al., 2020). Using this approach, we found that *Ctcf* was one of the overrepresented motifs in *Chd4* cKO RPCs. This suggests that the loss of *Chd4* might lead to increased recruitment of *Ctcf* to sites that are typically inaccessible, resulting in disorganization of genome looping as previously shown in cerebellar granule cells (Goodman et al., 2020).

The single cell analysis shows that there seems to be a striking effect on the UMAP plot of the

mutant in a nicely controlled experiment, but it is a little hard to understand what the take home is as all of the cells seem to be shifted but there are no clear differences in cell population numbers noted. The authors targeted an early postnatal timepoint to better understand *Chd4* gene function at the time it is acting, but the difficulty in assessing cell types because of ambiguities in identity make it difficult to interpret. It is not clear if there are detectable differences in the numbers of specific cell types such as rods or RPCs. Can cell percentages for these cells be included so as to observe whether there are changes in rod formation?

We added the cell proportion data for each replicate to the supplemental section of the paper (new Fig. S13). Based on our manual cell counts, the P1 scRNA-seq data are generated *prior* to the overt changes in the cell type composition of the retina, except for the expansion of RGCs. However, in agreement with other studies that used dissociated cells rather than nuclei, we captured very few RGCs. The dataset contains only 22 RGCs, although all 22 originated from the cKO replicates. In the scRNA-seq data, we do not observe changes in rod production in agreement with *Otx2* counts performed at P0. However, we do see the significant upregulation of many genes associated with the RPC fate, and we see downregulation of some neurogenic genes - notably including *Ascl1*, but also including *Neurod4* (*Math3*). We also saw an elevation in GO terms associated with apoptosis. We now include a GO terms analysis for upregulated DEGs in Fig. 6J.

We have also reordered the paper so that the reader can fully appreciate the changes in cell composition that occur at later stages, prior to the presentation of the scRNA-seq data. We hope that this will improve clarity.

One of the retina biology take-homes of the study is that *Chd4* regulates RPC to rod transitions in the postnatal period. The study does provide some solid evidence that there are indeed effects on RPCs (like developmental persistence past their normal time window) and also on rod formation (less rods formed). But because the NuRD complex could be working with many different transcription factors and involved in separate transcriptional events, it may be that the effects are not tied mechanistically together. Perhaps *Chd4* is involved in repressing the RPC state through effects on specific genes, but maybe it separately also keeps some genes turned off in rods that are toxic to them. The authors present a reasonable model and should keep it as a model - but without a deeper mechanistic understanding of how *Chd4* is exerting these effects it would be best to discuss other possibilities for the phenotypes that are observed.

As mentioned above, we now discuss the idea that rods are being killed rather than being underproduced. We have also added additional analyses on P5 stages in order to gain further insight into lineage termination in the *Chd4* cKO. We also revised the summary figure and legend (Fig. 8) in order to make clear that the observed changes in cell type composition can be driven by apoptosis, and that the proposed explanation of lineage changes are a model.

Minor Suggestions

It would help to have line numbers for reviewing.

Thanks for pointing this out. We have added line numbers in accordance with the required formatting for *Development*.

The original *Chx10* transgenic as described in Rowan and Cepko 2004 refers to the encoded fluorescent reporter as EGFP, but this manuscript refers consistently to it as EYFP. If the authors have a justification for this reference they should make the case, but if not, EGFP should be used.

We have now relabelled all figures to indicate that the green channel is "GFP". In all cases, we used epifluorescence to obtain these images, except for EdU stainings, where we performed IHC using a GFP antibody.

In Fig S1 the single cell plots don't make it obvious to a reader the expression differences of the *Chd3-5* genes. For example, neurogenic cells don't look that different in terms of shading compared to the other cell populations, but the boxplot looks very different. Inclusion of some other gene plots may help to make a comparison here or perhaps the scaling can be adjusted to make the differences and the general expression more obvious?

We tried a few different things to try to improve this figure, but unfortunately, we were unable to generate new plots from Clark et al. dataset directly using Scanpy due to technical reasons. We generated the images in the previous version of the figure using a web portal. Ultimately, we performed levels adjustments to the UMAP plots to better illustrate the signal from the genes, and we carefully matched the adjustments to the scale bars. Since the bar plots from the Clark et al. dataset were not robustly detecting these genes, we replaced them with data from our own scRNA-seq dataset. We now present violin plots for all of the major cell types from control cells at P1. We note that additional plots of *Chd3/4/5* from our own dataset are already presented in the paper in Figs. 6C, F and Fig. S14. We hope that these changes have addressed the issues flagged by the Reviewer above.

In Figure 2A some indication of the indicated Chd4 band size based on a ladder would be important to show that this is likely the predicted Chd4 band (and an indication of what that predicted band size is).

This is a very good point, since the Chd4 western revealed some weaker bands that did not disappear in the cKO sample. We have therefore added an image of the size marker to the western. In the text, we now state (pp. 6):

Full-length Chd4 protein (expected size 219 kDa) was efficiently abrogated in cKO retinas (Figure 2A-C).

We think that the observed band is consistent with the expected size.

In figure 2, is the Chx10 channel [which should be labeled EGFP or GFP^{Cre} (to represent the fusion protein) to be more direct in what is being observed] being shown in the Chd4 Hoechst picture? The retina presumably lacks the GFP signal because this animal doesn't have the transgene, but would be nice to show it here, or if the green channel is being shown to note that it is being visualized.

We think the Reviewer is referring to Fig. 2B. In this panel, the animal in question was Cre-negative. When performing the microscopy, we always inspect sections to look for GFP epifluorescence, but we found that the GFP channel was not acquired in the original image.

Consistency in labeling and description of the actual target that is being observed should be done when visualizing the GFPChx10iresCre transgene (GFP fluorescence, GFP antibody, Cre, Chx10 antibody). In Figure 2 "Chx10 is used", in figure 3 it is "Cre" to "Chx10 Cre" in fig s4- if they are in fact different this should be noted. A consistent method for labeling should be applied for all the figures that assess a marker expressed by the transgene.

We thank the Reviewer for pointing this out. We have re-labelled all figures to state "GFP". All relevant images were obtained via epifluorescence, except in EdU co-stainings, where we used an anti-GFP antibody.

The authors indicate that they did not observe any mosaicism as is commonly seen for the Chx10 transgene - it could be useful to show a whole retina view to show this.

We included images of whole retinal sections at E16.5 and P0 in Fig. S3, S8, and new Fig. S10, although the latter is at P15 where mosaicism (in bipolars) is not usually seen. We also removed the statement that mosaicism "was relatively minor" in our experiments, and added a new statement about mosaicism into the Methods section (pp. 19).

We avoided retinal fields that exhibited significant *Chx10-Cre-GFP* mosaicism as determined by GFP epifluorescence. We found that mosaicism in progenitors could be directly visualized up to P8, but could not be determined at later stages when GFP became expressed in bipolar cells. We therefore imaged retinal fields at random in P15 experiments.

We have done our utmost best to try to address potential mosaicism in our experiments, but do not wish to suggest that mosaicism was non-existent. We think that the prior version of the text implied this too strongly, and apologize for the confusion.

Later on when discussing the single cell data there is a reference to a possible mosaicism in one of the samples, so an image could bolster the case that there isn't. Though further information should be provided for the sample in the single cell experiment as noted in another part of this review.

We added a figure that illustrates the cells from each scRNA-seq replicate. There was definitely extensive mosaicism in cKO3, where about 50% of the cells segregated with controls clusters, while the other 50% segregated with the novel cKO clusters. We also present a dotplot in the figure that shows that *Chd4* mRNA expression is less reduced in cKO3, and that DEGs that upregulated in the cKO, such as *Chd5* and *Tcf15*, are less upregulated in this sample relative to the other two cKO replicates.

The Sox2 staining in Fig 7I and J looks like a lot of it is something else and not actually Sox2 which should be nuclear and is the band of cells at the top of the INL. The very strong Sox2 signal in cells outside of the MG layer and with cytoplasmic staining is not consistent with these being Müller glia and yet seems to be increasing in the mutant. This may be throwing off some of the counts of actual Müller cells - were they counted? Can another more specific marker be used? Perhaps sox9, p27, or Lhx2?

We wish to affirm that the Sox2 signal (now in Fig. 5L-N) is indeed nuclear. We apologize for the small panels that make such an evaluation difficult. We now include higher magnification images of Sox2 stainings in Fig. S9. As shown better in the new figure, the cells flagged by the reviewer have highly radially polarized nuclei that are very distinct in comparison to the amacrine nuclei. In all of our counts, we observe GFP signal (in non-amacrines) that overlaps perfectly with the Sox2 signal (as does hoechst staining).

We also tried Sox9 stainings, but found that the antibody that we purchased marked additional cells in the INL as well as Müllers. We therefore focused on *Rlb1*, which is a Müller-specific marker, but unfortunately, not a nuclear marker, making it very difficult to use for cell counting.

The viral tracing does not seem to have the sample size to be able to conclude statistical significance - the authors don't do so, but it is best to either increase the sample size to reach this threshold or not include the data.

As mentioned above, we elected to remove this experiment. We apologize for this issue.

For the single cell analysis there is a mention that "...these changes are observed despite probable mosaicism in one of the *Chd4* cKO samples...". It is not clear what is being referred to here. This point should be expanded as it would be important to know what the indication of this potential confounding effect is and what is its origin. Related is that it would be beneficial to show the individual UMAPs of each biological sample as its somewhat difficult to see each one - this could be done in supplemental material. It would also be useful to note the sex of each mouse, which could presumably be determined from gene expression such as *xist*.

We added a supplemental figure (Fig. S13) that presents the requested data.

In our analyses, we did not perform any comparisons between specific clusters that we think could potentially be undermined by mosaicism. Instead, we used unsupervised methods to annotate the cell types, and then performed pseudobulk analyses on a cell-type by cell-type basis, or across the whole population (i.e. only separating the cells by genotype).

If we were to compare the novel clusters obtained in the *Chd4* cKO samples to matched clusters present in the control replicates, we would presumably observe much stronger transcriptional effects, but we were not sure how to match each novel (i.e. cKO-specific) Leiden cluster with a specific matching control cluster perfectly. To avoid making errors in such assignments, we decided to err on the side of caution and not attempt to use UMAP clustering to power our comparisons. Instead, we relied only on genotype, which is read out independently of any clustering method by specific multi-seq barcodes.

The multi-seq approach allowed us to directly label each replicate with a barcode, obviating the

need to use clustering approaches to separate control versus cKO cells. For these reasons, we do not think that the mosaicism present in the cKO sample creates any confounds, although it presumably reduces the fold-changes seen in our DEGs, since the cKO replicates ostensibly include some cells that have not excised the *Chd4-Flox* cassette.

Thanks again.

References

- Brzezinski, J. A. t., Lamba, D. A. and Reh, T. A. (2010). Blimp1 controls photoreceptor versus bipolar cell fate choice during retinal development. *Development* **137**, 619-629.
- Dyer, M. A. and Cepko, C. L. (2000). p57(Kip2) regulates progenitor cell proliferation and amacrine interneuron development in the mouse retina. *Development* **127**, 3593-3605.
- MacDonald, R. B., Randlett, O., Oswald, J., Yoshimatsu, T., Franze, K. and Harris, W. A. (2015). Muller glia provide essential tensile strength to the developing retina. *The Journal of cell biology* **210**, 1075-1083.
- Poche, R. A., Furuta, Y., Chaboissier, M. C., Schedl, A. and Behringer, R. R. (2008). Sox9 is expressed in mouse multipotent retinal progenitor cells and functions in Muller glial cell development. *J Comp Neurol* **510**, 237-250.

Second decision letter

MS ID#: dev.204697R1

MS Title: Chd4 remodels chromatin to control retinal cell type specification and lineage termination

Authors: Sujay Shah; Suma Mediseti; José Alex Lourenço Fernandes; Pierre Mattar
Article Type: Research Article

Dear Dr Mattar,

I am delighted to tell you that your manuscript has been accepted for publication in *Development*, pending our standard publication integrity checks.

Reviewer 1

The authors were especially responsive to the reviewers' concerns. Their revised manuscript is greatly improved and will have a significant impact on the field.

Minor: check the y axis label in figure 4E (YFP vs GFP).

Reviewer 2

The authors have adequately addressed all of the comments that were made on the original submission.